

# How dry was the Younger Dryas? Evidence from a coupled $\delta^2$H-$\delta^{18}$O biomarker paleohygrometer, applied to the Lake Gemündener Maar sediments, Western Eifel, Germany

Johannes Hepp[a,b,#,1], Lorenz Wüthrich[c], Tobias Bromm[b], Marcel Bliedtner[c,2], Imke Kathrin

Schäfer[c], Bruno Glaser[b], Kazimierz Rozanski[d], Frank Sirocko[e], Roland Zech[c,2], Michael

Zech[b,f,3]

*aChair of Geomorphology and BayCEER, University of Bayreuth, Universitätsstrasse 30, D-95440 Bayreuth, Germany*

*bInstitute of Agronomy and Nutritional Sciences, Soil Biogeochemistry, Martin-Luther-*

*University Halle-Wittenberg, Von-Seckendorff-Platz 3, D-06120 Halle, Germany*

*cInstitute of Geography and Oeschger Center for Climate Change Research, University of Bern, Hallerstrasse 12, CH-3012 Bern, Switzerland*

*dFaculty of Physics and Applied Computer Science, AGH University of Science and Technology, Al. Mickiewicza 30, 30-059 Kraków, Poland*

*eInstitute of Geosciences, Group of Climate and Sediments, Johannes Gutenberg University of Mainz, J.-J.-Becher-Weg 21, D-55128 Mainz, Germany*

*fInstitute of Geography, Chair of Landscape- and Geoecology, Dresden University of Technology, Helmholtzstrasse 10, D-01062 Dresden, Germany*

*# corresponding author: johannes-hepp@gmx.de*

[1]Present Address: *Chair of Geomorphology and BayCEER, University of Bayreuth, Universitätsstrasse 30, D-95440 Bayreuth, Germany*
[2]Present Address: *Institute of Geography, Chair of Physical Geography, Friedrich-Schiller University of Jena, Löbdergraben 32, D-07743 Jena, Germany*
[3]Present Address: *Institute of Geography, Chair of Landscape- and Geoecology, Dresden University of Technology, Helmholtzstrasse 10, D-01062 Dresden, Germany*



**Abstract**

The Late Glacial to Early Holocene transition phase and particularly the Younger Dryas period, i.e. the major last cold spell in Central Europe during the Late Glacial, are considered crucial for understanding rapid natural climate change in the past. The sediments from Maar lakes in the Eifel, Germany, have turned out to be valuable archives for recording such paleoenvironmental changes.

For this study, we investigated a Late Glacial to Early Holocene sediment core that was retrieved from Lake Gemündener Maar in the Western Eifel, Germany. We analysed the hydrogen ($\delta^2$H) and oxygen ($\delta^{18}$O) stable isotope composition of leaf wax-derived lipid biomarkers (*n*-alkanes $C_{27}$ and $C_{29}$) and hemicellulose-derived sugar biomarkers (arabinose), respectively. Both $\delta^2$H and $\delta^{18}$O are suggested to reflect mainly leaf water of vegetation growing in the catchment of the Gemündener Maar. This enables the coupling of the results via a $\delta^2$H-$\delta^{18}$O biomarker paleohygrometer approach and allows calculating past relative air humidity values, which is the major advantage of the applied approach. Fundamental was the finding that the isotopic enrichment of leaf water due to evapotranspiration depends mainly on relative humidity. We hence use the coupled $\delta^2$H-$\delta^{18}$O biomarker approach to reconstruct the deuterium-excess of leaf water and in turn relative air humidity values corresponding to the vegetation period and daytime ($RH_{dv}$).

Most importantly, the results of the coupled $\delta^2$H-$\delta^{18}$O biomarker paleohygrometer approach (i) support a two-phasing of the Younger Dryas, i.e. a relative wet phase (on Allerød level) followed by a drier Younger Dryas ending, (ii) do not corroborate overall drier climatic conditions characterising the Younger Dryas or a two-phasing with regard to a first dry and cold Younger Dryas phase followed by a warmer period along with increasing precipitation amounts, and (iii) suggest that the amplitude of $RH_{dv}$ changes during the Early Holocene was more pronounced compared to the Younger Dryas. One possible driver for the unexpected Lake Gemündener Maar $RH_{dv}$ variations could be the solar activity.



## 1 Introduction

In order to evaluate the relevance of man-made climate change in the future, it is of great importance to study and understand large and rapid climate fluctuations in the past. Many studies have focused on the Late Glacial to Early Holocene transition phase, a period with various expressions in temperature, atmospheric circulation and hydrology worldwide (Alley, 2000; Brauer et al., 2008; Denton et al., 2010; Partin et al., 2015; Wagner et al., 1999). Particularly the Younger Dryas period, i.e. the major last cold spell in Central Europe during the Late Glacial just before the onset of the Holocene warm period (Denton et al., 2010; Heiri et al., 2014; Isarin and Bohncke, 1999), has long been considered crucial for understanding rapid natural climate change in the past (Alley, 2000). The sediments from maar lakes in the Eifel, Germany, have turned out to be valuable archives for paleoenvironmental reconstructions providing high resolution palynological, sedimentological and geochemical records for climate, vegetation and landscape history (Brauer et al., 2008; Brunck et al., 2015; Litt et al., 2003; Litt and Stebich, 1999; Sirocko et al., 2013; Zolitschka, 1998).

Lacustrine sedimentary lipid biomarkers, originating either from leaf waxes of higher terrestrial plants (Eglinton and Hamilton, 1967) or from aquatic organisms (Volkman et al., 1998), and especially their hydrogen isotope composition ($\delta^2 H_{\text{n-alkane}}$), are widely accepted as paleoclimate proxies (Huang et al., 2004; Mügler et al., 2008; Sachse et al., 2004; Sauer et al., 2001) demonstrating that the hydrogen isotope composition of the leaf waxes is well correlated with hydrogen isotope composition of precipitation ($\delta^2 H_{\text{prec}}$) (Hou et al., 2008; Rao et al., 2009). This confirms the great potential to establish records of the stable isotope composition of past precipitation (Sachse et al., 2012), similar to the well-known ice-core records from Greenland and from speleothems (Alley, 2000; Luetscher et al., 2015; Rasmussen et al., 2014), which among others provide information about changes of surface air temperature in the past (Dansgaard, 1964; Rozanski et al., 1993). However, the alteration of precipitation stable isotope signal either through leaf or lake water enrichment can challenge a direct reconstruction of $\delta^2 H_{\text{prec}}$ based on $\delta^2 H_{\text{n-alkane}}$ alone (Kahmen et al., 2013a; Mügler et al., 2008; Sachse et al., 2012). Apart from $\delta^2 H_{\text{n-alkane}}$, the oxygen isotope composition of hemicellulose/polysaccharide-derived sugars ($\delta^{18} O_{\text{sugar}}$) was established as a tool in paleoclimate research during the last years (Zech et al., 2011, 2013a, 2014a). Analogous to $\delta^2 H_{\text{n-alkane}}$, $\delta^{18} O_{\text{sugar}}$ is affected by the isotope composition of source water, which is closely related to the local precipitation ($\delta^{18} O_{\text{prec}}$), as well as by evapotranspirative $^{18}O$ enrichment (Tuthorn et al., 2014; Zech et al., 2013b, 2014b). However, the overall challenge of disentangling between the source water signal and changes caused by evapotranspiration can be potentially resolved by using a coupled $\delta^2 H$-$\delta^{18} O$ approach



(Henderson et al., 2010; Hepp et al., 2015, 2017; Tuthorn et al., 2015; Voelker et al., 2014, 2015; Zech et al., 2013a). While Voelker et al. (2014) presented a framework for using $\delta^2H$ and $\delta^{18}O$ of tree-ring cellulose in order to infer relative air humidity, Tuthorn et al. (2015) validated a coupled $\delta^2H$-$\delta^{18}O$ approach using leaf wax-derived $n$-alkanes and hemicellulose-derived

sugars, showing the potential to reconstruct mean relative air humidity changes along an Argentinian topsoil transect. Both approaches were successfully applied to loess-paleosol sequences (Hepp et al., 2017; Zech et al., 2013a) and sub-fossil wood (Voelker et al., 2015), whereas the $\delta^2H$-$\delta^{18}O$ coupling using terrestrial biomarkers in lacustrine archives is still missing. Indeed, already Sternberg (1988) suggested the coupling of aquatic $\delta^2H$ of lipids and

$\delta^{18}O$ cellulose in order to tangle lake water isotope composition more robustly, due to challenges of using $\delta^2H$ values of nitro-cellulose derived from lake sediments. Hence, coupled $\delta^2H$-$\delta^{18}O$ biomarker approaches applied to lacustrine archives were used to decipher between lake water evaporation history and changes in the isotope composition of the lake water input (Henderson et al., 2010; Hepp et al., 2015), based on a combination of $\delta^2H$ of palmitic acid and

$\delta^{18}O$ of carbonate as well as $\delta^2H_{n\text{-alkane}}$ and $\delta^{18}O_{sugar}$ records. Deriving quantitative relative humidity values is challenging without additional hydrological input parameters, due to past lake mass balance changes (e.g. Wolfe et al., 2001). However, quantitative paleoclimate changes from biomarker-isotope proxy data, like a relative humidity record, may be useful to assess atmospheric circulation pattern changes in the past (Voelker et al., 2015), which was so

far challenging due to a lack of applicable quantitative model approaches (Feng et al., 2007; Rach et al., 2014). Terrestrial $n$-alkane and sugar biomarker-based isotope records from Lake Gemündener Maar sediments, which are coupled in a $\delta^2H$-$\delta^{18}O$ paleohygrometer approach, have the potential to fill this gap by deriving a more robust relative air humidity record for Central Europe for the Late Glacial and Early Holocene transition.

25       This study was conducted based on the following objectives: (i) source identification of the sedimentary organic matter and the investigated $n$-alkanes and sugars (aquatic vs. terrestrial), (ii) reconstructing leaf water isotope composition based on compound-specific $\delta^2H$ and $\delta^{18}O$ values of the $n$-alkane and sugar biomarkers, (iii) reconstructing relative air humidity via the coupled $\delta^2H_{n\text{-alkane}}$-$\delta^{18}O_{sugar}$ paleohygrometer approach and (iv) inferring new

implications for middle European paleoclimate history from the established Lake Gemündener Maar relative air humidity record for the Late Glacial and Early Holocene transition. Overall, the potentials and limitations of the coupled $\delta^2H_{n\text{-alkane}}$-$\delta^{18}O_{sugar}$ paleohygrometer approach applied to lacustrine archive will be discussed.




## 2 Material & Methods

### 2.1 Lake Gemündener Maar and sampling

Lake Gemündener Maar is located in the Eifel volcanic fields in western Germany at an altitude of 407 m a.s.l (50°10'39.853"N, 6°50'12.912"E; Fig. 1A and B; Sirocko et al., 2013).

The Maar was formed during a phreatomagmatic explosion within the local Devonian siltstone (greywacke) around 20-25 ka BP (Büchel, 1993). The lake is 39 m deep at maximum and has a diameter of roughly 300 m. Due to its formation conditions the lake is almost circular with a lake surface area of 75,000 m$^2$ and is surrounded by a small catchment (Fig. 1B), with an area of 430,000 m$^2$ (Scharf and Menn, 1992). The lake is fed by precipitation and groundwater (no

surface inflow present). The catchment area is furthermore steep and densely vegetated with broad-leaved trees (Fig. 1C). The investigated samples were taken from the 8 m Gemündener Maar core (GM1), which was taken at approximately 20 m water depth near the center of the maar (Fig. 1B) with a Livingston piston corer (UWITEC, Mondsee, Austria). The core is part of the Eifel Laminated Sediment Archive Project of the Institute for Geoscience at Johannes

Gutenberg University Mainz (Sirocko et al., 2013, 2016).

(Fig. 1)

### 2.2 Bulk analysis and age control

Total carbon (TC), total organic nitrogen (TON), bulk carbon isotope composition

($\delta^{13}C_{TC}$) and nitrogen isotope composition ($\delta^{15}N$) were determined at the Institute of Agronomy and Nutritional Sciences, Soil Biogeochemistry, Martin-Luther-University Halle-Wittenberg, using an EuroVector EA 3000 elemental analyzer (Hekatech, Wegberg, Germany) coupled via a Conflo III Interface to a Delta V Advantage isotope ratio mass spectrometer (IRMS; both from Thermo Fisher Scientific, Bremen, Germany). Additionally, total organic carbon (TOC)

and bulk $\delta^{13}C$ of the total organic carbon ($\delta^{13}C_{TOC}$) were assessed after removal of carbonate with 32% HCl fumigation followed by a neutralization step with moist NaOH, both for 24 h under 60°C and vacuum conditions. This allows calculating carbon to nitrogen ratio based on total organic contents (TOC/TON). Laboratory standards from the International Atomic Energy Agency (IAEA) as well as from United States Geological Survey (USGS) with known total

carbon, nitrogen, $^{13}C$ and $^{15}N$ contents (IAEA N2, IAEA CH6, IAEA NO3, IAEA CH7, IAEA 305A, USGS 41) were used for calibration. The $^{13}C$ and $^{15}N$ contents are expressed in the common δ-notation as relative to an international standard ($\delta^{13}C$: Vienna Pee Dee Belemnite, VPDB; $\delta^{15}N$: atmospheric N$_2$, Air). It should be noted that TON and $^{15}N$ contents are only reported for measurements prior to carbonate removal. This is based on Walthert et al. (2010)



and Harris et al. (2001), who reported lower N contents and slightly more positive $\delta^{15}N$ after HCl fumigation, respectively. Bulk analyses were carried out on 112 samples, covering a section of 606 cm to 727 cm depth of the Lake Gemündener Maar GM1 core.

The investigated sediments are partially laminated and the Laacher See Tephra (673 to
680 cm core depth; Fig. 2A) is clearly visible. The latter can be used as chronological marker due to the varve counted age of 12,880 a BP in the adjacent Meerfelder Maar (Fig. 2C; Brauer et al., 1999). The second tie point to establish a chronology for the Lake Gemündener Maar core is a radiocarbon-dated piece of charcoal in 727 cm core depth, which dated to 13,800 ± 110 cal a BP (Fig. 2C). This age is derived from a $^{14}C$ age of 11,950 ± 65 a BP as part of the
supplement material of Sirocko et al. (2013), calibrated using CalPal-software (Weninger and Jöris, 2008) calculated with the IntCal13 calibration curve (Reimer et al., 2013). The uncertainty of the calibrated $^{14}C$ age represents the 68% probability range. The onset of the Holocene in the Greenland NGRIP core, dated to 11,650 a BP (Walker et al., 2009), is used as wiggle match due to the distinct TOC increase in 642 cm depth of the Gemündener Maar core
(Fig. 2A, C and D). The Late Glacial to Holocene transition is commonly well recorded in maar sediments from the Eifel region, i.e. clear changes in deposition as well as pollen pattern, varve counted to 11,600 a BP in Lake Holzmaar (Zolitschka, 1998) and to 11,590 a BP in Lake Meerfelder Maar (Brauer et al., 1999; Litt et al., 2001, 2003; Litt and Stebich, 1999). Additionally, the clear appearance of *Corylus* Pollen in 620 cm depth (Fig. 2B; derived from
unpublished pollen analysis by F. Sirocko) was wiggle matched to the appearance of *Corylus* pollen in the adjacent Lake Holzmaar core, dated to 10,450 a BP (Sirocko et al., 2016). The beginning of the Younger Dryas period was set to 12,680 a BP (200 a after the Laacher See Tephra), according to the findings from Lake Meerfelder Maar (Litt et al., 2001). The investigated core section from 607 cm to 694 cm depth, with regard to the biomarkers, covers
the time between ~ 13,200 a and 9,700 a BP, i.e. the Allerød, Younger Dryas, Preboreal, and the beginning of the Boreal (Fig. 2C). Assuming a constant sedimentation rate between the markers results in an average resolution of 58 a/cm; the minimum and maximum resolution are 19 and 164 a/cm, respectively. The part above the Laacher See Tephra reveals a lower mean resolution (63 a/cm) than the section below (30 a/cm).

## 2.3 Biomarker and compound-specific isotope analysis

For $\delta^2H$ analyses of *n*-alkanes as well as $\delta^{18}O$ analyses of sugars, 59 samples were prepared from 607 cm to 694 cm depth of the Lake Gemündener Maar GM1 core (Fig. 3A and



B), in order to cover the core section with already high TOC content and the Late Glacial to Holocene transition (Fig. 2A, C and D). *n*-Alkanes were extracted from 1-6 g freeze-dried and grinded samples by microwave extraction at 100°C for 1 h, using 15 ml of solvent (dichloromethane and methanol, in a ratio 9:1). The resultant total lipid extracts were separated

over aminopropyl silica gel (Supelco 45 μm) filled pipette columns. Nonpolar compounds (including *n*-alkanes) were eluted with *n*-hexane. The fraction was spiked with a known amount of 5α-androstanone, used as internal standard. Identification and quantification was carried out on an Agilent MS 5975 (EI) interfaced with an Agilent 7890 GC equipped with a 30 m fused silica capillary column (HP5-MS 0.25 mm i.d., 0.25 μm film thickness), and a split/splitless

injector operating in splitless mode at 320°C. Carrier gas was helium and the temperature program was 1 min at 50°C, from 50 to 200°C at 30°C min$^{-1}$, from 200 to 320°C at 7°C min$^{-1}$, 5 min at 320°C. Data recording comprised the total ion count (scan mode from m/z 40 to m/z 600) and single ion monitoring (m/z 57, 71, 85 and 99). Concentrations were calculated relative to the internal standard and to an external standard (*n*-$C_{21}$ to *n*-$C_{40}$ alkane mixture, Supelco),

injected in different concentrations (40 ng/μl, 4 ng/μl, 1 ng/μl, 0.4 ng/μl).

   Prior to compound-specific isotope analyses, the *n*-alkanes were further purified. The nonpolar fractions were passed over a pipette column filled with activated $AgNO_3$ impregnated silica gel and a pipette column filled with zeolite (Geokleen). After drying, the zeolite was removed using hydrofluoric acid and the *n*-alkanes were recovered by liquid-liquid extraction

with hexane. The purified *n*-alkane fractions were measured for their compound-specific stable hydrogen isotope composition ($\delta^2$H). The measurements were performed at the Institute of Geography, University of Bern on an IsoPrime 100 IRMS, coupled to an Agilent 7890A GC via a GC5 pyrolysis/combustion interface operating in pyrolysis modus with a Cr (ChromeHD) reactor at 1000°C. Samples were injected with a split/splitless injector. The GC was equipped

with 30 m fused silica column (HP5-MS, 0.32 mm inner diameter, 0.25 μm film thickness). The precision was checked by co-analyzing a standard alkane mixture (*n*-$C_{27}$, *n*-$C_{29}$, *n*-$C_{33}$) with known isotope composition (A. Schimmelmann, University of Indiana), injected twice every six runs. The samples were analyzed in threefold repetitions (except from the samples in 622 and 672 cm depth), and the analytical precision was generally better than 5‰. The stable

hydrogen isotope compositions are given in the δ-notation ($\delta^2$H$_{n\text{-alkane}}$) versus Vienna Standard Mean Ocean Water (VSMOW). The $H_3^+$-correction factor was checked every two days and stayed stable over the course of measurements at 3.14. The $\delta^2$H$_{n\text{-alkane}}$ values refer to the area weighted mean of the $\delta^2$H values of *n*-alkanes with 27 and 29 carbon atoms (*n*-$C_{27}$, *n*-$C_{29}$), respectively, because of their relatively high abundance in the samples.



The sample preparation for $\delta^{18}O$ analyses of hemicellulose/polysaccharide-derived sugars followed standard procedures at the Institute of Agronomy and Nutritional Sciences, Soil Biogeochemistry, Martin-Luther-University Halle-Wittenberg, according the method of Zech and Glaser (2009). The monosaccharide sugars were hydrolytically extracted from

samples containing approximately 10 mg total organic carbon with 10 ml of 4 M trifluoroacetic acid at 105°C for 4 h, applying the method described by Amelung et al. (1996). After filtration over glass fibre filters, the extracted sugars were cleaned using XAD-7 (to remove humic-like substances) and Dowex 50WX8 columns (to remove interfering cations). Afterwards, the purified samples were freeze-dried and derivatized by adding methylboronic acid (4 mg in 400

µl pyridine) for 1 h at 60°C (Knapp, 1979). The methylboronic acid derivatization method ensures that the investigated pentoses arabinose and xylose as well as the deoxyhexoses fucose and rhamnose yield only one peak in the chromatograms (Gross and Glaser, 2004).

The compound-specific $\delta^{18}O$ measurements were performed using a Trace GC 2000 coupled to a Delta V Advantage IRMS via an $^{18}O$-pyrolysis reactor (GC IsoLink) and a ConFlo

IV interface (all devices from Thermo Fisher Scientific, Bremen, Germany). Each sample was measured in threefold repetition, embedded in-between co-derivatized sugar standards in various concentrations and known $\delta^{18}O$ values. The $\delta^{18}O$ values of the samples are expressed in δ-notation ($\delta^{18}O_{sugar}$) versus the Vienna Standard Mean Ocean Water (VSMOW). The measured $\delta^{18}O_{sugar}$ values were corrected for drift, amount and area dependency and also for

the hydrolytically introduced oxygen atoms that form carbonyl groups with the C1 atoms of the sugar molecules (Zech and Glaser, 2009). Mean standard uncertainties for the triplicate measurements of all 59 samples are 0.6‰, 0.7‰ and 0.7‰ for arabinose, fucose and xylose, respectively. The $\delta^{18}O_{sugar}$ values refer to the $\delta^{18}O$ values of the monosaccharides arabinose, fucose and xylose. Rhamnose areas, respectively concentrations, were too low for reliable

isotope measurements in many samples.

## 2.4 Conceptual framework for coupling $\delta^2H_{n\text{-alkane}}$ with $\delta^{18}O_{sugar}$ results

The coupled $\delta^2H_{n\text{-alkane}}$-$\delta^{18}O_{sugar}$ approach was described in detail by Tuthorn et al. (2015) and Zech et al. (2013a). The most fundamental assumption of the approach is that the

isotope composition of leaf water can be reconstructed by applying biosynthetic fractionation factors on the measured $\delta^2H_{n\text{-alkane}}$ and $\delta^{18}O_{sugar}$ values (Fig. 4). The concept is furthermore based on the observation that isotope composition of global precipitation plots typically close to the global meteoric water line (GMWL; $\delta^2H_{prec} = 8 \cdot \delta^{18}O_{prec} + 10$; Dansgaard, 1964). In





Germany, a local meteoric water line (LMWL$_{Germany}$) slightly deviating from GMWL was described by Stumpp et al. (2014) ($\delta^2$H$_{prec}$ = 7.72 ± 0.13 · $\delta^{18}$O$_{prec}$ + 4.90 ± 0.01; Fig. 4), which we used as the baseline for our calculations. The quite similar LMWLs for Trier ($\delta^2$H$_{prec}$ = 7.81 ± 0.08 · $\delta^{18}$O$_{prec}$ + 5.06 ± 0.60) and Koblenz ($\delta^2$H$_{prec}$ = 7.80 ± 0.07 · $\delta^{18}$O$_{prec}$ + 2.68 ± 0.53) as well as the GMWL are additionally displayed in Fig. 4 for comparison. The local precipitation is the source for soil water and shallow groundwater, which in turn acts as source water for plants. During daytime, however, leaf water is typically enriched compared to the source water due to evapotranspiration trough the stomata, plotting therefore right of the GMWL and the LMWLs (Fig. 4; Allison et al., 1985; Bariac et al., 1994; Walker and Brunel, 1990). The leaf water reservoir at the evaporative sites achieves quickly steady-state conditions (Allison et al., 1985; Bariac et al., 1994; Gat et al., 2007; Walker and Brunel, 1990), meaning that the isotope composition of the transpired water vapor is equal to the isotope composition of the source water utilized by the plants during the transpiration process. The evaporative enrichment of leaf water under steady-state conditions, can be described via a Craig-Gordon model (e.g. Flanagan et al., 1991; Roden and Ehleringer, 1999) by the following expression (e.g. Barbour et al., 2004):

$$\delta_e \approx \delta_s + \varepsilon^* + \varepsilon_k + (\delta_a - \delta_s - \varepsilon_k)\, \frac{e_a}{e_i}, \qquad\qquad \text{(Eq. 1)}$$

where $\delta_e$, $\delta_s$ and $\delta_a$ are the hydrogen and oxygen isotope compositions of leaf water at the evaporative sites, source water and atmospheric water vapor, respectively, $\varepsilon^*$ are the equilibrium enrichment expressed as $(1-1/\alpha_{L/V}) \cdot 10^3$ where $\alpha_{L/V}$ is the equilibrium fractionation between liquid and vapor in ‰, $\varepsilon_k$ are the kinetic fractionation parameters for water vapor diffusion from intracellular air space through the stomata and the boundary layer, both for $^2$H and $^{18}$O, respectively, and $e_a/e_i$ is the ratio of atmospheric vapor pressure to intracellular vapor pressure. When leaf temperature is equal to air temperature, the $e_a/e_i$ ratio represents the relative humidity of the local atmosphere (RH). If the plant source water and the local atmospheric water vapor are in isotopic equilibrium, the term $\delta_a - \delta_s$ can approximated by $-\varepsilon^*$. Thus. Eq. (1) can be reduced to:

$$\delta_e \approx \delta_s + \left(\varepsilon^* + \varepsilon_k\right) (1 - \text{RH}). \qquad\qquad \text{(Eq. 2)}$$

The kinetic fractionation parameters ($\varepsilon_k$) are typically related to stomatal and boundary layer resistances with respect to water flux (Farquhar et al., 1989). Since direct measurements of those plant physiological parameters can be hardly assessed in a paleo application we used the kinetic enrichment parameters $C_k$ instead, derived from a more generalized form of the Craig-





Gordon model, for describing the kinetic isotope enrichment for $^2$H and $^{18}$O, respectively, which leads to Eq. (3) (Craig and Gordon, 1965; Gat and Bowser, 1991):

$$\delta_e \approx \delta_s + \left(\varepsilon^* + C_k\right)(1 - RH).\qquad\text{(Eq. 3)}$$

In a $\delta^2$H-$\delta^{18}$O diagram, the hydrogen and oxygen isotope composition of leaf and source water can be described as a local deuterium-excess (d) = $\delta^2$H - 7.72 · $\delta^{18}$O (Stumpp et al., 2014) in one equation by using the slope of the LMWL$_{Germany}$ (Eq. 4). This approach is comparable to the deuterium-excess definition from Dansgaard (1964), who used the equation d = $\delta^2$H - 8 · $\delta^{18}$O for a measure of the parallel deviation between a given point in the $\delta^2$H-$\delta^{18}$O diagram to the GMWL.

$$d_e \approx d_s + \left(\varepsilon_2^* - 7.72 \cdot \varepsilon_{18}^* + C_k^2 - 7.72 \cdot C_k^{18}\right)(1 - RH)\qquad\text{(Eq. 4)}$$

Where $d_e$ and $d_s$ are the deuterium-excess values of the leaf water at the evaporative sites and the source water, respectively, and the equilibrium ($\varepsilon_2^*$ and $\varepsilon_{18}^*$) and kinetic enrichment parameters ($C_k^2$ and $C_k^{18}$) are expressed for both isotopes. From Eq. (1) to Eq. (4) the primary control of the relative humidity on the isotope composition of the leaf water is demonstrated when stomata are open through transpiration. If $d_e$ can be derived from compound specific $\delta^2$H and $\delta^{18}$O measurements of the *n*-alkane and sugar biomarkers, which derive $\delta^2$H$_e$ and $\delta^{18}$O$_e$ values for the purpose of calculating $d_e$ values via the equation $d_e = \delta^2$H$_e$ - 7.72 · $\delta^{18}$O$_e$, and the $d_s$ can be approximated also from the deuterium-excess of the LMWL$_{Germany}$ (= 4.9). Accordingly, Eq. (4) can be rearranged in order to calculate relative humidity of the local atmosphere normalized to leaf temperature as given by Eq. (5) (Hepp et al., 2017; Tuthorn et al., 2015; Zech et al., 2013a):

$$RH \approx 1 - \frac{\Delta d}{\left(\varepsilon_2^* - 7.72 \cdot \varepsilon_{18}^* + C_k^2 - 7.72 \cdot C_k^{18}\right)},\qquad\text{(Eq. 5)}$$

where $\Delta d$ is the distance between $d_e$ and $d_s$, calculated as $\Delta d = d_e - d_s$. Equilibrium fractionation parameters ($\varepsilon_2^*$ and $\varepsilon_{18}^*$) are derived from empirical equations of Horita and Wesolowski (1994) with mean daytime growth-period temperature of 14.8°C (from 6 a.m. to 7 p.m. and April to October, derived from the Nürburg-Barweiler station, approx. 25 km northeast of Lake GM; hourly data from 1995 to 2015 from Deutscher Wetterdienst, 2016). Equilibrium fractionation factors equal 83.8 and 10.15‰ for $^2$H and $^{18}$O, respectively. The kinetic fractionation parameters ($C_k^2$ and $C_k^{18}$) for $^2$H and $^{18}$O are set to 25.1 and 28.5‰, respectively, according to Merlivat (1978), who reported maximum values during the molecular diffusion process of water through a stagnant boundary layer. The assumption that maximum kinetic fractionation occurs seems to be most suitable for sedimentological application where a signal averaging over



decades can be assumed (see above and discussion in Zech et al., 2013a). It should be also noted that $\varepsilon_k$ values of broadleaf trees and shrubs over broad climatic conditions are well in the range with used $C_k^2$ and $C_k^{18}$ values, revealing 23.9 ($\pm$0.9) and 26.7‰ ($\pm$1.0) for $\delta^2H$ and $\delta^{18}O$, respectively (derived from supplementary data of Cernusak et al., 2016).

The nominator of Eq. (5) describes the parallel distance between the deuterium-excesses of LMWL and leaf water at the evaporative sites which is converted into relative humidity values, while the denominator is a combination of the slopes of LMWL and LEL. This means in turn that the quantification with Eq. (5) is done by obtaining the distance between the source water points, calculated via the intersects between the individual local evaporation lines and the

LMWL$_{Germany}$, and the leaf water points. The underlying slope of those LELs can be derived from Eq. (6) via the Craig-Gordon model using the same assumptions as outlined above in a rearranged form (Eq. 6; Zech et al., 2013a). When using the fractionation parameters from above, the slope of the LEL is constant over time, independent from RH and equal to ~ 2.8 (Eq. 6).

$$S_{LEL} = \frac{\delta_e^2 - \delta_s^2}{\delta_e^{18} - \delta_s^{18}} \approx \frac{\varepsilon_2^* + C_k^2}{\varepsilon_{18}^* + C_k^{18}} \qquad \text{(Eq. 6)}$$

In order to provide an uncertainty interval in terms of measurement precision covering the Lake Gemündener Maar relative humidity record, pooled standard errors (SE) of $d_e$ values were used (according to Eq. 7) to generate maximum and minimum values for Eq. (5), which result in a lower and upper relative humidity limit.

$$SE\ d_e = \sqrt{\left(SE\ \delta^2H_{n\text{-alkane}}\right)^2 + 7.72 \cdot \left(SE\ \delta^{18}O_{sugar}\right)^2} \qquad \text{(Eq. 7)}$$

**2.5 Monte-Carlo-based correlation analysis**

      In order to account for uncertainties regarding the here established Lake Gemündener Maar Late Glacial to Early Holocene age-depth model (cf. section 2.2), we applied a Monte-Carlo-based correlation analysis to compare more robustly our RH$_{dv}$ record with the IntCal13 $^{14}$C production rates (data from Muscheler et al., 2014). This is assessed by varying the onset

of *Corylus*, the onset of the Holocene and the charcoal $^{14}$C age of the Lake Gemündener Maar core. The Laacher See Tephra is accurately varve counted with an error of $\pm$ 40 a (Brauer et al., 1999), which is negligible small compared to the others, and therefore excluded from the variation procedure. The onset of *Corylus* pollen is dated to 10,450 a BP in ~ 8.4 m depth





(Sirocko et al., 2016) in the adjacent Lake Holzmaar core. To put an error on this date, four [14]C ages derived from twig material of Lake Holzmaar core are used (from the supplements of Sirocko et al. 2013) two above the *Corylus* onset (4.8 m depth: $3560 \pm 25$ and $3615 \pm 30$ a BP) and two below (8.5 m depth: $9740 \pm 140$ and $9670 \pm 90$ a BP). By using the CalPal-software

(Weninger and Jöris, 2008) with the IntCal13 calibration curve (Reimer et al., 2013) these ages were calibrated to $3860 \pm 30$, $3930 \pm 40$, $11080 \pm 220$ and $11000 \pm 160$ a cal BP. The errors of the calibrated [14]C ages refer to the 68% probability range. This gives a mean error for the upper dates of $\pm 35$ a, and $\pm 190$ a for the lower ones. In error propagation we combined both mean errors to $\pm 225$ a for the onset of *Corylus* pollen in the Lake Gemündener Maar core. For the

onset of the Holocene in the Lake Gemündener Maar sediments, which was defined as wiggle match between the TOC increase in 642 cm depth and the onset of the Holocene in the Greenland NGRIP core (dated to 11,650 a BP Walker et al., 2009), we combined the maximum counting error for the Holocene onset in the NGRIP core of $\pm 99$ a (Walker et al., 2009) together with a potential time lag of 170 a (Rach et al., 2014) to $\pm 269$ a. Such a time lag was detected

in sedimentary hydrogen isotope values of aquatic and terrestrial *n*-alkanes from Lake Meerfelder Maar concerning a lag in hydrologic and environmental effects in Western Europe occurring 170 a later compared to Greenland cooling at the GI-1 to GS-1 transition (Rach et al., 2014). Finally, the derived errors for the onset of the Holocene ($\pm 269$ a), the onset of *Corylus* ($\pm 225$ a) and the calibrated charcoal [14]C age ($\pm 110$ a) were used in a correlation procedure based

on a Monte-Carlo-Simulations (realized in R, version 3.2.2, R Core Team, 2015). In a first mode the time steps were set to 5 a, while the refining, 10 a around the highest correlation coefficient, was done in yearly resolution. This procedure allows deriving optimized ages for those three dates, which in turn enables the calculation of an adjusted age-depth model.

## 3 Results & Discussion

### 3.1 Source identification of bulk organic matter and of the investigated *n*-alkane and sugar biomarkers

For basic sedimentological characterization, measurements of TC and TOC (both Fig. 2D), TN (Fig. 2E), $\delta^{15}N$ (Fig. 2F), $\delta^{13}C_{TC}$ and $\delta^{13}C_{TOC}$ (both Fig. 2G) as well as the TOC/TN

ratio (Fig. 2H) are displayed from 605 cm to 727 cm depth. TC and TOC values range from 0.6 to 20.7% and from 0.6 to 19.7%, respectively. Slightly higher TC values compared to TOC can be attributed to the occurrence of carbonate. TN reveals a range of 0.1 to 1.4%. In general, TOC and TN depth profiles show the same trends (TOC vs. TN: $r = 0.99$, $p < 0.001$, $n = 110$), i.e. an increase during the Allerød, lower values during the Younger Dryas, and larger variations



during the Preboreal and the Boreal. The $\delta^{15}N$ values range from 0 to 5‰, showing the maximum and minimum within the Allerød period. $\delta^{13}C_{TC}$ and $\delta^{13}C_{TOC}$ reveal values between -31 to -17‰ and -36 to -24‰, respectively. While $\delta^{13}C_{TC}$ show maximum values in 703 cm depth, $\delta^{13}C_{TOC}$ is decreasing continuously from the beginning to the end of the Allerød,

followed by increasing values during the Younger Dryas and the Preboreal/Boreal, interrupted by a short decrease around the beginning of the Holocene. $\delta^{13}C_{TC}$ clearly show the presence of carbonate between 690 cm and 727 cm depth, due to the more positive $\delta^{13}C_{TC}$ values compared to $\delta^{13}C_{TOC}$ values, which is however not clear noticeable from the difference between TC and TOC (compare Fig. 2G with 2D). TOC/TN ratios range from 5 to 16 with the end of the Allerød

revealing increasing ratios, while the late Younger Dryas shows slightly decreasing ratios and the Preboreal is marked by the highest ratios.

(Fig. 2)

The source of organic matter in lacustrine sediments of small lakes, as one of the most crucial questions and challenges when interpreting organic proxies from lacustrine sedimentary

records (Meyers and Ishiwatari, 1993), can either be autochthon (aquatic origin) or allochthon (terrestrial origin). The carbon to nitrogen ratio (C/N) and $\delta^{13}C$ values are most common proxies for sedimentary source determination. While non-vascular aquatic organisms often reveal C/N ratios between 4 and 10 (due to low amounts of cellulose and lignin), vascular plants show commonly C/N ratios of 20 and higher (Meyers and Ishiwatari, 1993). Already Prahl et al.

(1980) used a C/N ratio of > 12 as threshold for a dominance of terrestrial input (Fig. 2H). The Lake Gemündener Maar samples below 642 cm (onset of the Holocene) show TOC/TN ratios of mostly < 12, pointing to a high aquatic input, whereas in the section above the ratios shift to values > 12, suggesting a partial contribution of terrestrial material (Fig. 2H; Meyers and Ishiwatari, 1993). Lake Gemündener Maar $\delta^{13}C_{TOC}$ values (Fig. 2G) are well in range with $C_3$

land plants and lacustrine algae (Meyers and Lallier-Vergés, 1999). Most likely, no changes between $C_3$ and $C_4$ land plants occurred in Lake Gemündener Maar catchment history, due to the absence of $C_4$ land plants. Thus, no clear additional information about the sedimentary organic matter origin in Lake Gemündener Maar sediments can be achieved neither by $\delta^{13}C_{TOC}$ alone (Lücke et al., 2003), nor by combining $\delta^{13}C_{TOC}$ with TOC/TN ratios (Meyers and Lallier-

Vergés, 1999). When considering that both $\delta^{13}C_{TOC}$ and C/N values of terrestrial organic matter are additionally affected by mineralization and degradation, resulting in more positive $\delta^{13}C_{TOC}$ values and lower C/N ratios (e.g. Zech et al., 2007), a straightforward interpretation of those proxies seems to be challenging. Much less common, $\delta^{15}N$ values are used to derive information about sedimentary organic matter origin (Meyers and Ishiwatari, 1993). Source determination



using $^{15}$N is based on the finding that dissolved nitrate, which is the nitrogen source for aquatic plants, and atmospheric nitrogen, which is the nitrogen source for land plants, largely differ in their isotope ratios (Peterson et al., 1985). For Lake Gemündener Maar, $\delta^{15}$N values (Fig. 2F) imply more aquatic input below 690 cm depth while above stronger terrestrial imprint is

obvious which is in principle resembling the TOC/TN record. However, numerous alternation pathways of sedimentary $^{15}$N due to biogeochemical processes, like nitrogen uptake by plants, various nitrogen sources, discrimination during denitrification, and diagenesis, complicate the use of $\delta^{15}$N as direct source determination proxy (Altabet et al., 1995; Meyers and Lallier-Vergés, 1999; Wolfe et al., 1999). However, the $\delta^{15}$N peak in 690 cm depth coincides with a

striking brownish layer (Fig. 2A), revealing low C/N ratios. Possible this marks the input of soil material from the catchment, with soils typically being more positive in $\delta^{15}$N compared to plant material (Natelhoffer and Fry, 1988). Since the origin of the sedimentary organic matter cannot be determined exactly, the Lake Gemündener Maar sedimentary TOC values (Fig. 2D) should also not be over interpreted. Higher TOC contents during the Allerød, Preboreal and

Boreal are typically linked to warmer conditions favorable for aquatic biomass production, whereas the lower TOC values during the Younger Dryas are associated with colder and dryer climate with increasing minerogenic sedimentation (e.g. Brauer et al., 1999). Still, the low TOC content during the late Allerød section (below 690 cm) of Lake Gemündener Maar point to a low organic matter production in the lake as well as allochtonous organic matter input. Also the

Allerød sediments of Lake Holzmaar are not characterized by TOC maxima (Lücke et al., 2003).

Despite the uncertainties describing the source of the bulk sedimentary organic matter in Lake Gemündener Maar sediments using bulk characteristics, the origin of the sedimentary biomarkers, namely $n$-alkanes and sugars, needs to be addressed. This is important because

aquatic biomarkers incorporate the isotope composition of lake water, while terrestrial biomarkers incorporate the isotope composition of leaf water (Huang et al., 2004; Kahmen et al., 2013a; Mügler et al., 2008; Sachse et al., 2004, 2012; Sauer et al., 2001; Tuthorn et al., 2014; Zech et al., 2013b, 2014b). For a successful application of the coupled $\delta^2$H$_{n\text{-alkane}}$-$\delta^{18}$O$_{sugar}$ approach, in terms of deriving relative humidity values, the biomarkers have to originate from

leaf material of higher terrestrial plants (Hepp et al., 2017; Tuthorn et al., 2015; Zech et al., 2013a). Regarding the $n$-alkane biomarker, high amounts of the chain lengths $n$-C$_{27}$ and $n$-C$_{29}$ were detected in the Lake Gemündener Maar sediments, which enables then robust compound-specific $\delta^2$H measurements afterwards. Those homologues originate typically from epicuticular leaf wax layers of higher terrestrial plants in the catchment (e.g. Eglinton and Hamilton, 1967).





The investigated sugar biomarker pattern were previously studied by Hepp et al. (2016), showing that arabinose is primarily of terrestrial origin (higher vascular plants), whereas fucose and xylose are mostly of aquatic origin (algae). This interpretation was in agreement with (paleo)soil and sediment data compiled from the literature (cf. references in Hepp et al., 2016).

Finally, it should be clarified if the investigated sedimentary biomarkers are primarily of leaf origin from higher plants in the catchment. While this holds most likely true for the investigated $n$-alkanes, due to their primary source from epicuticular leaf wax layers of higher terrestrial plants, the neutral sugar arabinose is found in leaf, stem and moss material (Jia et al., 2008; Prietzel et al., 2013; Zech et al., 2012, 2014b). Firstly, high coverage of the catchment

by trees is obvious (Fig. 1C), leading to high degree of leaf litter input in autumn. Secondly, vegetation grown on the maar crater wall is supposed to be the main input of biomass into the lake, which is typical for steeply shored mare lakes, while the shoreline vegetation contribution should be comparable small. Thirdly, an arboreal to nonarboreal pollen ratio of ~ 84, as mean value throughout the investigated core section is reported (derived from unpublished pollen

analysis by F. Sirocko). We therefore assume that arabinose, as well as $n$-$C_{27}$ and $n$-$C_{29}$, originate primarily from hemicellulose structures and from epicuticular waxes of tree leaf material, respectively.

## 3.2 Reconstructing leaf water isotope composition based on $\delta^2H_{n\text{-alkane}}$ and
$\delta^{18}O_{sugar}$

The $\delta^2H$ depth profiles reveal variations of -222 to -134‰ and -220 to -147‰ for $n$-$C_{27}$ and $n$-$C_{29}$, respectively (Fig. 3A). Their $\delta^2H$ patterns correlate highly significant with each other (r = 0.7, p < 0.001, n = 59). Weighted mean $\delta^2H$ values were calculated using the relative amounts of $n$-$C_{27}$ and $n$-$C_{29}$. More negative $\delta^2H$ values are obvious during the Younger Dryas

(~ 17‰ more negative, with a mean of -194‰), compared to the Allerød (-182‰), Preboreal (-178‰) and Boreal (-171‰), with regard to the weighted mean $n$-alkane depth profile. Overall the Holocene part of the core average at ~ -175‰, where consistently also largest fluctuations are observed (~50‰). The $\delta^{18}O$ values for arabinose, xylose and fucose range from 28 to 41‰, 26 to 45‰ and 27 to 46‰, respectively (Fig. 3B). They overall reveal similar trends (arabinose

vs. xylose: r = 0.7, p < 0.001, n = 59; arabinose vs. fucose: r = 0.8, p < 0.001, n = 59; xylose vs. fucose: r = 0.8, p < 0.001, n = 59). All sugar records show a clear shift to more positive values at the Younger Dryas to Holocene transition. While xylose and fucose exhibit a change of ~ 8 and 6‰, arabinose $\delta^{18}O$ values show a less positive shift of ~ 3‰ (changes are based on the





mean $\delta^{18}O$ values for the Younger Dryas compared to the Preboreal/Boreal period). Xylose is however slightly more negative throughout the Allerød and Younger Dryas compared to arabinose and fucose. Consistently less pronounced changes can be observed for the Allerød-Younger Dryas transition of 1.8, 1.4 and 0.5‰ for xylose, fucose and arabinose, respectively

(based on the mean $\delta^{18}O$ values for the Allerød compared to the Younger Dryas). A distinct minimum during the early Preboreal (633 cm depth) characterizes all three $\delta^{18}O$ sugar records.

(Fig. 3)

When applying the $\delta^2H_{n\text{-alkane}}$-$\delta^{18}O_{sugar}$ paleohygrometer approach to lacustrine sediments, not only the basic origins of the sedimentary biomarkers have to be clarified (cf.

section 3.1), but also prerequisite that leaf water isotope composition is recorded in the biomarkers and can thus be reconstructed from them. The isotope composition of leaf wax $n$-alkanes and leaf (hemi-)celluloses from higher plants are known to be strongly related to the water in which they are biosynthesized. They reflect basically the isotope composition of leaf water during photosynthetic activity (Barbour and Farquhar, 2000; Cernusak et al., 2005;

Kahmen et al., 2013a; Sachse et al., 2012). Hence, the isotope signature of the paleo leaf water, $\delta^{18}O_l$ and $\delta^2H_l$, respectively, can be reconstructed by using known biosynthetic fractionation factors (Fig. 4; Eq. 8 and 9). For this purpose, fractionation factors of $-160‰$ for $^2H$ of alkanes $n$-$C_{27}$ and $n$-$C_{29}$ ($\varepsilon^2_{bio}$; Sachse et al., 2012; Sessions et al., 1999), and $+27‰$ for $^{18}O$ of hemicellulose sugar arabinose ($\varepsilon^{18}_{bio}$; Cernusak et al., 2003; Schmidt et al., 2001; Sternberg et

al., 1986; Yakir and DeNiro, 1990) are assumed.

$$\delta^{18}O_l = (\delta^{18}O_{arabinose} - \varepsilon^{18}_{bio})/(1 + \varepsilon^{18}_{bio}/1000) \qquad (\text{Eq. 8})$$

$$\delta^2H_l = (\delta^2H_{n\text{-alkane}} - \varepsilon^2_{bio})/(1 + \varepsilon^2_{bio}/1000) \qquad (\text{Eq. 9})$$

Sucrose exported from photosynthesizing leaves was shown to be $\sim +27‰$ more positive compared to leaf water (e.g. Cernusak et al., 2003). However, also the cellulose biosynthesis is associated with an enrichment of $\sim +27‰$ compared to the synthesis water, as shown in growth experiments (e.g. Sternberg et al., 1986; Yakir and DeNiro, 1990). This means

that the isotope signal from the leaf water incorporated in the transport sugar sucrose can be dampened by oxygen exchange with local synthesis water during (hemi-)cellulose biosynthesis in the sink tissue (e.g. Barbour and Farquhar, 2000). Barbour and Farquhar (2000) related this signal damping to the proportion of unenriched source water contributing to the local synthesis water ($p_x$) and to the proportion of exchangeable oxygen during cellulose synthesis ($p_{ex}$). Latter

is often assumed to be rather constant around 0.40, as estimated from leaf and wood cellulose of *Eucalyptus globulus* and values compiled from literature (Cernusak et al., 2005), meaning





that 40% of the oxygens in the stem cellulose exchanged. Indeed, studies on leaf material of *Eucalyptus globulus* (Cernusak et al., 2005) and for *Gossypium hirsutum* (Barbour and Farquhar, 2000) report a total damping of 38% during leaf cellulose synthesis. However, already Wang et al. (1998) presented a large range of damping between 4 and 100% for leaf

cellulose synthesis (provided in the supplementary material of Cheesman and Cernusak, 2017), based on plant data grown under the same conditions at Jerusalem Botanical Gardens. The signal damping depends also on $p_x$. Since no direct $p_x$ measurements on dicotyledonous plant leaves are available so far (Liu et al., 2017), no clear conclusion can be drawn about the oxygen signal damping during (hemi-)cellulose synthesis in the leaves of dicotyledonous plants. The

signal damping effect described for cellulose synthesis, however, could partially be lost during synthesis of leaf hemicellulose structures. Pentoses, like the hemicellulose-derived arabinose, are biosynthesized via decarboxylation of the carbon at position six (C6) from glucose (Altermatt and Neish, 1956; Burget et al., 2003; Harper and Bar-Peled, 2002). Waterhouse et al. (2013) showed that the oxygens at C6 position in glucose moieties are most strongly affected

by the exchange with local water medium of 80%, as indicated by heterotrophic cellulose synthesis. This leads to the approximation that the influence of the source water and the respective oxygen exchange can be considered negligible for leaf arabinose from trees. This is supported by a strong relationship between leaf water and leaf hemicellulose $\delta^{18}O$ values of dicotyledonous plants grown in climate chambers, pointing to a strong incorporation of the leaf

water signal into the leaf hemicellulose sugars (Hepp et al., 2018). The reduction of the signal damping does not seem to apply for the respective stem hemicelluloses, with a mean $^{18}O$ signal damping of $\sim$ 66% (Sternberg, 2014; Zech et al., 2014c) For a $C_4$ grass *Cleistogenes squarrosa*, Liu et al. (2016) presented a signal damping range between 34 and 53%. Direct $p_x$ measurements in the grass leaf growth-and-differentiation zone of the same $C_4$ grass reveal

values between 0.9 and 1.0 for $^{18}O$ (Liu et al., 2017). Comparison of grasses to the tree stem regarding the signal transfer from leaf water into the (hemi-)cellulose was already reported by Helliker and Ehleringer (2002). A correction seems to be required against the background of *Poaceae* pollen concentrations between 11 and 33% during the Allerød and Younger Dryas periods in Lake Gemündener Maar sediments (Fig. 3C; derived from unpublished pollen

analysis by F. Sirocko). A correction procedure based on mass balance considerations is given in Eq. (10) in order to adjust $\delta^{18}O_l$ to $\delta^{18}O_l^{\#}$:

$$\delta^{18}O_l^{\#} = \{(\delta^{18}O_l - \delta^{18}O_s)/[f_{non\text{-}grasses} + (1 - 0.4) - f_{non\text{-}grasses} \cdot (1 - 0.4)]\} + \delta^{18}O_s. \qquad \text{(Eq. 10)}$$

The correction presented in Eq. (10) is based on assumptions that 40% (0.4) of the leaf water enrichment is lost during hemicellulose biosynthesis of grass leaves, which is well in range with




values presented in the literature for cellulose synthesis in tree rings and grasses, respectively (Cernusak et al., 2005; Liu et al., 2016). Furthermore, the *Poaceae* pollen concentration in percentage is used to calculate the fraction of non-grassy pollen [$f_{non-grasses}$ = (100 – *Poaceae*)/100] corresponding to the non-grassy biomarker contribution, which may serve as

rough approximation. For a paleo application, $\delta^{18}O_s$ remains a priori unknown. Therefore, the intercept between the individual LEL´s (Eq. 6) and the LMWL of Germany were used to generate $\delta^{18}O_s$ values. The overall assumption is still that (hemi-)cellulose is solely biosynthesized from immediate products of photosynthesis. Storage substances like starch indeed contribute to leaf cellulose synthesis (Terwilliger et al., 2001). So far it can only be

speculated that this does not significantly influence the strong relationship between (hemi-)cellulose and leaf water (Terwilliger et al., 2002), as appropriate studies are lacking. Leaf water inhomogeneity and sucrose synthesis gradients within leaves could weaken the leaf water to (hemi-)cellulose correlation (Lehmann et al., 2017; Santrucek et al., 2007). Additionally, changes in $^{18}O$ fractionation ($\varepsilon^{18}_{bio}$) could hamper a robust leaf water reconstruction based on

leaf hemicellulose. Sternberg and Ellsworth (2011) showed for heterotrophically generated cellulose a strong temperature dependence of the biosynthetic fractionation factor. A temperature dependency was also found for $\varepsilon^{18}_{bio}$ of stem hemicelluloses from dicotyledonous plants, recalculated by Sternberg (2014) based on results from Zech et al. (2014a). In the reply the authors show that rather the $^{18}O$ leaf water signal damping ($p_{ex}$), associated with the steam

hemicelluloses synthesis, can be related to the temperature growth (Zech et al., 2014c). However, the evaluation of leaf material from the same plants strongly suggests an $^{18}O$ fractionation factor of around +27‰ (Hepp et al., 2018). This confirms the validity of Eq. (8) in order to derive leaf water from arabinose $\delta^{18}O$ values.

The climate chamber experiment of Hepp et al. (2018) revealed furthermore a $^2H$

fractionation between long chained *n*-alkanes (*n*-$C_{29}$ and *n*-$C_{31}$) and leaf water around -156‰. This is well in line with the commonly reported −160‰ (Sachse et al., 2012; Sessions et al., 1999) and is confirmed by climate chamber studies of dicotyledonous plants (Kahmen et al., 2011b, 2013a; Tipple et al., 2015). However, the latter studies reveal also a range of ~ 35‰, interpreted as species-specific effects during *n*-alkane biosynthesis. Much more pronounced is

the difference between dicotyledonous and monocotyledonous $C_3$ plants regarding the degree to which the leaf water isotope enrichment is transferred into leaf *n*-alkanes (Gamarra et al., 2016; Kahmen et al., 2013a). While dicotyledonous plants show signal transfer rates of 96% on average (Kahmen et al., 2013a), a larger range between 38 and 61% are found for monocotyledonous plants (Gamarra et al., 2016). The latter imply that 39 to 62% of the $^2H$ leaf





water enrichment is not recorded by the $n$-alkanes of grasses. The outlined discussion lead to the conclusion that $\delta^2H_l$ can be reconstructed based on $\delta^2H_{n\text{-alkane}}$ values (using $\delta^2H$ of $n$-C$_{27}$ and $n$-C$_{29}$ via Eq. 9). However, a correction seems to be required for grass $n$-alkane input, equivalent to Eq. (10):

$$\delta^2H_l^{\#} = \{(\delta^2H_l - \delta^2H_s)/[f_{non\text{-}grasses} + (1 - 0.5) - f_{non\text{-}grasses} \cdot (1 - 0.5)]\} + \delta^2H_s, \qquad \text{(Eq. 11)}$$

where $\delta^2H_l^*$ are the grass corrected $\delta^2H_l$ values. The $\delta^2H_s$ values and the non-grassy pollen fraction are defined as in Eq. (10). The mass balance correction presented in Eq. (11) is based on assumptions that only 50% of the leaf water enrichment is incorporated by the $n$-alkanes during biosynthesis in grass leaves (based on a mean value calculated from values reported by Gamarra et al., 2016). Besides the discussion about correct and constant biosynthetic

fractionation factors and signal damping factors during biosynthesis, the uncorrected $\delta^2H$ values of $n$-C$_{27}$ and $n$-C$_{29}$ can be compared to $\delta^2H$ results from $n$-C$_{29}$ alkanes extracted from close-by Meerfelder Maar sediments (Rach et al., 2014). This record covers a comparable timespan from 13,100 to 11,000 varve a BP. The Lake Gemündener Maar $\delta^2H_{n\text{-alkane}}$ record resembles the previously presented Lake Meerfelder Maar results with an overall range of 75‰ (13,200 to

9,700 a BP), with ~ -8‰ more negative values in mean during the Younger Dryas compared to the Allerød and a ~ +17‰ shift in average from Younger Dryas to Holocene. The Lake Meerfelder Maar $\delta^2H$ $n$-C$_{29}$ values, interpreted as reflecting mainly $\delta^2H$ composition of paleo leaf water, reveal a range of 68‰, ~ -8‰ more negative values in mean during the Younger Dryas compared to the Allerød and a Younger Dryas to Holocene shift in average of ~ +19‰

(Rach et al., 2014).

In summary, the above outlined discussion results in four different leaf water correction scenarios as input for Eq. (5) (Tab. 1).

(Tab. 1)

**3.3 Reconstructing relative humidity based on the coupled $\delta^2H_{n\text{-alkane}}$-$\delta^{18}O_{sugar}$ paleohygrometer approach**

The biomarker-based leaf water values ($\delta_l = \delta^2H_l$, $\delta^{18}O_l$ via Eq. 8 and 9) result in deuterium-excess values of leaf water ($d_l$), which range between -125 and -30‰ (Fig. 4 and 5A). A clear enrichment is observed compared to the source water signal ($d_s$), which was set to

+4.9‰ as derived from the deuterium-excess of the LMWL$_{Germany}$ (cf. section 2.4; Fig. 4).

(Fig 4)

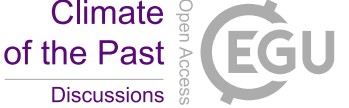



Using $d_l$ values (and $d_s$) as input for Eq. (5), relative humidity values during daytime and vegetation period ($RH_{dv}$) can be calculated (scenario 1 in Tab. 1). Reconstructed $RH_{dv}$ values reveal a range from 32 to 82% (Fig. 5B). The error bars covering the Lake Gemündener Maar $RH_{dv}$ record, calculated using pooled $d_e$ standard errors ranging from 3.2 to 44.4‰ according

Eq. (7), results in a span of 1.7 to 23.4% in relative humidity terms. It should be noted that for two data points (622 and 672 cm depth) only one, instead of three, $\delta^2H_{n\text{-alkane}}$ measurement result is available, avoiding the calculation of pooled $d_e$ standard errors for those. However, these two data points were excluded from the ranges given above. The $RH_{dv}$ record shows distinctly lower relative humidity values during the late Younger Dryas period, early and late

Preboreal, while the Allerød, first half of the Younger Dryas, middle Preboreal and the Boreal are characterized by more humid conditions. The plausibility of the reconstructed $RH_{dv}$ values is furthermore confirmed by the comparison to modern $RH_{dv}$ values from the adjacent meteorological station Nürburg-Barweiler (approx. 25 km northeast of Lake GM; Fig. 5C). Mean reconstructed $RH_{dv}$ values (mean $RH_{dv}$ = 53%; mean $RH_{dv}$ upper limit = 46%; mean $RH_{dv}$

lower limit = 61%) resembles the modern mean vegetation period (growth time; April to October) daytime (6 a.m. to 7 p.m.) humidity value of 67% (hourly data from 1995 to 2015 from Deutscher Wetterdienst, 2016). In addition, the range of the reconstructed $RH_{dv}$ values of 50% is well in agreement with in modern $RH_{dv}$ variability of 45%, within a range of 48% to 93% (definition and meteorological station details as above). The availability of Lake

Gemündener Maar *Poaceae* pollen concentrations (Fig. 3C and 5C; derived from unpublished pollen analysis by F. Sirocko) enables calculating three correction leaf water scenarios, which take into account that grass-derived biomarkers are typically less sensitive leaf water recorders (scenarios 2-4 in Tab. 1; cf. 3.2). The effects on the deuterium-excess of leaf water as well as on the corresponding relative humidity values are shown in Fig. 5A and B.

(Fig. 5)

More specifically, the full correction for grass-derived biomarker contribution (scenario 4 in Tab. 1) result in 0.0 to -6.2 (mean -1.9%) when comparing $RH_{dv}$ to $RH_{dv}^{\#}*$. The corresponding $d_l^{\#}*$ value changes range between -11.8 and -0.1‰. Such small changes are still far below the pooled standard error introduced by the measurement errors (Fig. 5A and B). When only grass-

corrected arabinose-based $\delta^{18}O_l^{\#}$ values are used, whereas the *n*-alkane $\delta^2H_l$ input stays uncorrected (scenario 3 in Tab. 1), a range of -22.2 to -0.1‰ can be obtained in the $d_l^{\#}$ value differences, and the corresponding relative humidity value changes range between -0.1 and -11.6%, in terms of comparing $RH_{dv}$ to $RH_{dv}^{\#}$. The opposite scenario became obvious when just the *n*-alkane based grass corrected $\delta^2H_l*$ values are used to calculate relative humidity values





while arabinose based $\delta^{18}O_l$ remains uncorrected (scenario 2 in Tab. 1). This lead to shifts of 0.0 to 5.5%, when comparing $RH_{dv}$ to $RH_{dv}^*$. Indeed, the respective positive $d_l^*$ value changes of 0.1 to 10.4‰ are strongly associated to the *Poaceae* pollen concentration. It is therefore clear why the full corrected $RH_{dv}^{\#*}$ values differ only minimal from the uncorrected $RH_{dv}$. Based on

that finding, it could be argued that $RH_{dv}$ values derived from $\delta^2H_l$ and $\delta^{18}O_l$ are relatively robust against grassy biomarker contributions. Further implications against a correction arose from the fact that the relative humidity values derived from the coupled $\delta^2H_{n\text{-alkane}}$-$\delta^{18}O_{sugar}$ approach are interpreted as reflecting daytime and vegetation period relative humidity ($RH_{dv}$; Tuthorn et al., 2015). Leaf biomass production in a temperate forest reveals a bimodal

distribution regarding the height due to a dense understory vegetation and high leaf production of annually deciduous trees at ground and upper canopy level, respectively (Graham et al., 2014). Moreover, strong gradients in relative humidity can occur in temperate forest ranging from near to saturated conditions (> 95%) close to the ground (< 5 m height) and highly unsaturated conditions (down to ~ 60%) throughout the upper part of the forest (> 5 m height)

in a diurnal cycle (Parker, 1995). This suggest that tree leaves are in average build up under lower humidity conditions, compared to the understory (leaf) biomass. With regard to grass-derived *n*-alkane and sugar contributions to the Lake Gemündener Maar $\delta^2H$ and $\delta^{18}O$ biomarker records below 642 cm depth, this would suggest a higher impact of the more humid understory signal, what could lead to higher reconstructed relative humidity values when

*Poaceae* input is high. This would contradict the applied grass correction (Eq. 10 and 11), which should therefore be even more carefully reviewed in order to avoid an over interpretation of the corrected values.

It is widely accepted that the bulk leaf water is less enriched compared to the leaf water at the evaporative sites (e.g. Allison et al., 1985; Barbour, 2007). Furthermore the simplified

Craig-Gordon model to derive RH values (Eq. 5) requires per definition the input of the deuterium-excess of evaporative site leaf water, as outlined in section 2.4 (Flanagan et al., 1991; Roden and Ehleringer, 1999). In order to estimate the effect on the deuterium-excess of differences between Craig-Gordon predicted $\delta_e$ (evaporative site leaf water) and measured $\delta_l$ (bulk leaf water), deuterium-excess ($d_l$ and $d_e$) values were calculated for broadleaf trees and

shrubs using the supplementary dataset of Cernusak et al. (2016). Using the complete leaf water isotope data of that supplements, which are available for Australia (Kahmen et al., 2013b) and Hawaii (Kahmen et al., 2011a), a deuterium-excess formulation of $d = \delta^2H - 6.86 \cdot \delta^{18}O$, based on the additional available xylem water isotope data, was found. The thus calculated values average at -88.5 (±35) and -79.6‰ (±29) for $d_l$ and $d_e$, respectively. This means that in this

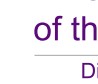
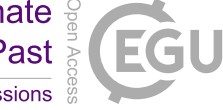

dataset, $d_l$ is in average 8.9‰ more negative than $d_e$, with a standard deviation of 16‰. Since it is general accepted that $\delta_e$ is more positive compared to $\delta_l$, one would expect $d_e$ to be more negative compared to $d_l$. Illustrated in a $\delta^2H$-$\delta^{18}O$ diagram, the $\delta_e$ should plot further right on a LEL, while the $\delta_l$ is assumed to be a mixture plotting therefore closer to the source water, still on the LEL. However, the average difference is still smaller than the mean analytical error of biomarker-derived $d_l$ with 15‰ (see Eq. 7 for calculation). A deuterium-excess correction to account for the difference between $d_l$ and $d_e$ seems therefore not crucial.

For a successful relative humidity reconstruction, a constant LEL slope is assumed, meaning isotopic equilibrium between plant source water and the local atmospheric water vapor. For giving a estimation weather this assumption is valid here, one should distinguish between isotopic equilibrium between local atmospheric water vapour and local precipitation and that between local atmospheric water vapour and local source water available for plants, while latter is of importance here. Under present-day European climate, close to isotope equilibrium conditions at ground-level temperature between monthly precipitation and atmospheric water vapour can be assumed (Araguás-Araguás et al., 2000; Jacob and Sonntag, 1991), with the exception of snow during winter months (snow preserves isotope signature established high in the clouds, at significantly lower temperatures). However, the source water for plants is mostly a mixture between soil water and shallow groundwater. The isotope composition of deeper soil water horizons and groundwater is constant and close to annual (weighted) mean isotope signature of precipitation (e.g. Herrmann et al., 1987). Therefore, during summer time, when the plants are photosynthetic active, plant source water is in isotope disequilibrium with atmospheric water vapour to which the leaves are exposed. The impact on the calculated RH values of the possible lack of equilibrium between plant source water (having the expected isotope signature equal to that of annual weighted mean of local precipitation) and the local atmospheric water vapour during vegetation period, was evaluated using the data provided by Jacob and Sonntag (1991), who measured isotope composition of precipitation and atmospheric water vapour in Heidelberg, Germany during the period 1981 to 1989. The mean difference between source water to water vapour equilibrium fractionation (= ε*) calculated for the vegetation period (April-October) and the apparent fractionation ($\varepsilon_{ap}$), derived from Jacob and Sonntag (1991) data, amounts to 18.3 and 1.57‰ for $^2H$ and $^{18}O$, respectively ($\varepsilon_{ap}$ is smaller than ε*). This apparent fractionation was then used in Eq. (1) instead of the difference $\delta_a$ - $\delta_s$ and RH values were recalculated. The effect of this replacement, however, lead to an average RH changes of only -1.7% (±0.9), which is far below the analytical errors of the deuterium-excess of leaf water. Apart from a potential disequilibrium between the atmospheric water vapor



and the source water, also variable air temperatures can alter the LEL slope. However, a temperature change of ± 10°C shift the reconstructed $RH_{dv}$ values just around ± 1%, via slightly different equilibrium fractionation parameters (leading to an LEL slope range from 3.0 to 2.6; cf. section 2.4). This temperature range largely exceed the expected variation throughout the

Late Glacial to Early Holocene transition, inferred e.g. from plant-based mean July temperatures from Isarin and Bohncke (1999). This is in line with the previous finding that such temperature changes are negligible for the relative humidity reconstruction compared to the measurement errors of biomarker isotope analyses (cf. Zech et al., 2013a, and discussion therein). Overall, the used LEL slope of 2.8 (Fig. 4), as well as the above presented range for

possibly temperature changes, are well in agreement with field and laboratory studies (Allison et al., 1985; Bariac et al., 1994; Gat et al., 2007; Tipple et al., 2013; Walker and Brunel, 1990).

       A further assumption of the presented paleohygrometer approach is that Δd are solely interpreted as the result of leaf water isotope changes and should thus reflect changes in leaf water evaporative enrichment (Hepp et al., 2017; Tuthorn et al., 2015; Zech et al., 2013a).

Indeed, the evaporative enrichment of leaf water is strongly driven by the relative air humidity in the direct surrounding of the leaf (e.g. Eq. 3; Cernusak et al., 2016). However, soil water enrichment is not accounted. Generally, soil water enrichment would shift the plant soil water along the soil evaporation line right of the $LWML_{Germany}$, for which our model does not account leading to an underestimation of RH if, in spite of this, the deuterium-excess of the

$LMWL_{Germany}$ approximates $d_s$. Soil water evaporation typically affects only the upper centimeters of the soil (~ 10 cm; Dubbert et al., 2013), while the trees utilize deeper and thus unenriched soil water (Dawson, 1993). An additional prerequisite is the stability of the deuterium-excess and slope of the $LMWL_{Germany}$ through the past. According to Stumpp et al. (2014), the long-term deuterium-excess of precipitation values from 28 sites do not show

pronounced relationships to local climate conditions of the site. All reported values are close to 10‰, which indicates that Atlantic air masses are the main moisture source for Germany (e.g. Rozanski et al., 1993). In addition, the deuterium-excess in local precipitation from the close-by stations Trier and Koblenz reveal small variability on monthly, annual and long-term basis compared to the pooled standard error of $d_l$, which averages at 15‰. For Trier monthly averages

(March to October) range from 5.3 to 8.7‰, annual precipitation weighted means from 1.9 to 10.6‰, and long-term precipitation weighted mean is 6.7‰ (±2.2); for Koblenz the values range between 2.1 and 6.4‰, 1.4 and 8.7‰, and the long-term weighted mean is 4.1‰ (±1.8) (derived from IAEA/WMO, 2018). Finally, deuterium-excess variability in Greenland and Antarctic ice-core does not exceed 4‰ over the here relevant time scale (Masson-Delmotte et





al., 2005; Stenni et al., 2010). In addition, paleowater samples from Europe suggest that the atmospheric circulation patterns over European continent were rather constant throughout the past 35,000 years, which also implies that long-term deuterium-excess of precipitation does not deviate substantially from the modern value (Rozanski, 1985). In summary, the variations in

the slope of the LMWL of Germany are assumed to be rather small over longer timescales. With regard to the detailed discussion in this section, the $RH_{dv}$ seems to be suitable for paleoclimate record comparison.

### 3.4 How dry was the Younger Dryas in Western Europe?

While it is well known that the Younger Dryas was a cold spell occurring in the Northern Hemisphere during the Late Glacial (Denton et al., 2010; Heiri et al., 2014; Isarin and Bohncke, 1999), there is much less clear evidence concerning moisture supply/availability and relative humidity changes during the Younger Dryas. The Lake Gemündener Maar $RH_{dv}$ results suggest relatively humid conditions at end of the Allerød and the first phase of the Younger Dryas (~

56%), which are in the same range as modern $RH_{dv}$ values (Fig. 5B; derived from the adjacent meteorological station Nürburg-Barweiler, approx. 25 km northeast of Lake GM; hourly data from 1995 to 2015 from Deutscher Wetterdienst, 2016). This is followed by a clear $RH_{dv}$ decrease of ~ 11% towards the end of the Younger Dryas, compared to the Allerød and first phase of the Younger Dryas (Fig. 6A). Such a two phasing of the Younger Dryas has been

suggested previously based on multiproxy climate data (Isarin et al., 1998). In more detail, Isarin et al. (1998) point to a subdivided Younger Dryas into a cold and humid first phase followed by drier and warmer conditions for western Europe. It is moreover speculated that a shift of the mean sea-ice margin during winter in the North Atlantic Ocean slightly to the north could have caused this two phasing. Reduced cyclonic activity and precipitation affected

thereby primarily Western Europe because this region was situated at the southern margin of the main storm tracks during the first Younger Dryas period (Isarin et al., 1998). The authors presented also evidence for a strengthening of the westerly winds in Western Europe as consequence of northwards shifted North Atlantic Ocean sea-ice margin during the late Younger Dryas period. However, the varve thickness record from Lake Meerfelder Maar

laminated sediments show a distinctive reduction of the sedimentation rates at 12,240 varve a BP, dividing the Younger Dryas in a first phase with thicker varves while the second period is characterized by slightly reduced sedimentation rate (Brauer et al., 1999). The thicker varves during the early Younger Dryas (between 12,680 and 12,240 varve a BP) are used along with geochemical analysis as indicator for stronger winter winds (Brauer et al., 2008). In line with



this, Brauer et al. (1999) interpreted high biogenic opal contents and *Prediastrum* remain concentrations during the early Younger Dryas as enhanced aquatic productivity due to an increased nutrient supply caused by soil erosion and the reworking of littoral sediments. The varve formation throughout the second Younger Dryas period (between 12,240 and 11,590 varve a BP) is characterized as mainly controlled by snowmelt driven surface runoff (Brauer et al., 1999). Moreover, the authors speculate if during that time the Meerbach creek came into connection with the Meerfelder Maar, which could be possibly linked to enhanced precipitation amounts. In summary, the interpretations derived from the Younger Dryas sediments of Lake Meefelder Maar (Brauer et al., 2008, 1999) is not in accordance with the results from Isarin et al. (1998), as well as the $RH_{dv}$ results presented here (Fig. 6A). Not fully consistent with both outcomes are the continuously dry conditions during the Younger Dryas, as derived from the difference between terrestrial- and aquatic-plant-derived *n*-alkane $\delta^2H$ values extracted from Lake Meefelder Maar sediments (Rach et al., 2014). It should be noted that Rach et al. (2017) used this difference as main input parameter in their dual-biomarker approach. In more detail, the ~ 10% drop of relative humidity changes between the Allerød and the Younger Dryas period is well comparable to the ~ 10% relative humidity difference between the Allerød and the second phase of the Younger Dryas at Lake Gemündener Maar $RH_{dv}$ record (comparisons is based on mean period values). Moreover, the regeneration towards Allerød humidity level around 12,000 a BP, the distinct rise at the end of the Younger Dryas as well as the overall variability obvious in Lake GM reconstructed $RH_{dv}$ values of 50% throughout the Late Glacial to Holocene transition fit well to the relative humidity changes from Lake Meerfelder Maar. Rach et al. (2017) reconstructed a comparable range of ~ 40% throughout the investigated time period covering the Younger Dryas (11,000 to 13,100 varve a BP). However, the continuous dry conditions during the Younger Dryas period reported by Rach et al. (2017, 2014) do not resemble the Lake Gemündener Maar $RH_{dv}$ record. Probably some greater extent of lake water enrichment during the late Allerød and the late Younger Dryas extended the difference between terrestrial and aquatic *n*-alkane $\delta^2H$ values, leading consequently to an underestimation of the dryness during those periods. Indeed, Lake Holzmaar, which seems to be comparable to Lake Meefelder Maar at least for the drainage conditions via one creek, shows a difference of 7.4‰ in $\delta^2H$ between inflow and lake water (Sachse et al., 2004). It is moreover shown by Sachse et al. (2004) that no significant correlation occurs between *n*-$C_{23}$ alkane $\delta^2H$ values, between lake water and precipitation along a European lake surface transect. Finally it can be questioned if Lake Meerfelder Maar *n*-$C_{23}$ alkanes solely originate from *Potamogeton* (Rach et al., 2014) with recent publications indicating that *n*-$C_{23}$ is not necessarily interpreted of aquatic origin (Aichner et al., 2018; Rao et al., 2014). Probably some extend could also originate from soil



microbes, transported into the lake sediments during the early Younger Dryas due to high erosion rates (Brauer et al., 1999). In summary, this adds uncertainty to the continuous dryness of the Younger Dryas based on the difference between terrestrial and aquatic $n$-alkane $\delta^2H$ values (Rach et al., 2014, 2017). Similarly, these uncertainties could potentially affect the

difference between terrestrial and aquatic $n$-alkane $\delta^2H$ values from Hässeldala Port Younger Dryas lake sediments from Southern Sweden (Muschitiello et al., 2015). The authors used the approach presented by Rach et al. (2014) with $\delta^2H$ values of $n$-$C_{21}$ representative for lake water, respectively summer precipitation, and $n$-$C_{27}$, $n$-$C_{29}$ and $n$-$C_{31}$ as reflecting leaf water enrichment. The calculated difference between terrestrial and aquatic $n$-alkane $\delta^2H$ values

suggest more humid conditions at the beginning of the Younger Dryas followed by a more or less steadily trend towards drier conditions, peaking around 11,700 a BP (Muschitiello et al., 2015). This would be, however, in line with Lake Gemündener Maar $RH_{dv}$ minimum between ~ 11,700 and 11,900 a BP, within age uncertainties of the Lake Gemündener Maar age-depth model. A recent publication from the Southern Pyrenees, analyzing triple oxygen and hydrogen

isotopes of gypsum to reconstruct relative humidity changes, also report more humid conditions at the beginning of the Younger Dryas compared to middle and the end of this phase (Gázquez et al., 2018).

(Fig. 6)

When searching for a possible driver of relative humidity changes during the Late

Glacial to Early Holocene transition, the Lake Gemündener Maar $RH_{dv}$ record can be compared to the $^{14}C$ production and $^{10}Be$ flux rates (Fig. 6B), derived from IntCal13 and the Greenland ice-cores (GRIP, GISP2), respectively (Muscheler et al., 2014). Such records are commonly known as solar activity (and thus insolation) proxies (Stuiver and Braziunas, 1988; Vonmoos et al., 2006). Our results suggest that the end of the Allerød and the first phase of the Younger

Dryas are characterized by relatively high $RH_{dv}$ and strong centennial-scale variability (Fig. 6A). This coincides with high and strongly variable $^{14}C$ production and $^{10}Be$ flux rates. The clear drop of the $RH_{dv}$ values towards the end of the Younger Dryas is also in agreement with a change to a higher solar activity level. Although $RH_{dv}$ initially recovered slightly at the onset of the Holocene, the Preboreal is mostly a phase of low $RH_{dv}$. Much higher variability is

pronounced than throughout the Younger Dryas. Lowest values are recorded during the late Preboreal (~ 10,600 a BP), which coincide with increased solar activity. However, a short (~300 a) and pronounced phase of very high $RH_{dv}$ occurred ~ 11,200 a BP. We dub this wet period the Preboreal Humid Phase, which is again accompanied by low solar activity (Fig. 6A and B). The Preboreal Humid Phase should not be confused with the Preboreal Oscillation (Björck et



al., 1997). The Preboreal Oscillation is a short cold event recorded in Greenland ice-cores ~11,400 ka (Rasmussen et al., 2007) and led to more arid conditions at least in the Netherlands according to palynological results (Bos et al., 2007; van der Plicht et al., 2004). These pollen records also show the existence of a pronounced humid phase thereafter, i.e. the Preboreal

Humid Phase. Widespread glacial advances in the Alps are also attributed to the Preboreal Oscillation (Moran et al., 2016), but given the dating uncertainties they may actually rather reflect increased precipitation during the Preboreal Humid Phase. Lake Gemündener Maar $RH_{dv}$ record is overall in good agreement with other local and regional paleoclimate reconstructions. The coherence between $RH_{dv}$ and solar activity, in particular during the

Younger Dryas and Preboreal Humid Phase, suggests a possible causality. It is widely accepted that the Younger Dryas and the Preboreal Oscillation are related to freshwater forcing in the North Atlantic (e.g. Fisher et al., 2002; Murton et al., 2010; Muschitiello et al., 2015). However, the causes and mechanisms responsible for climate and environmental changes during the rest of Holocene remain vague, and more research is needed to investigate the possible influence of

solar activity (Renssen et al., 2007; Rind, 2002). This can also be inferred from the conducted Monte-Carlo-Simulation based correlation procedure between Lake Gemündener Maar $RH_{dv}$ values and the IntCal13 $^{14}C$ production rates, which gives a maximum correlation coefficient of 0.37. As such, ~ 14% variability in the Lake Gemündener Maar $RH_{dv}$ record can be explained by solar activity changes. This is achieved when the $^{14}C$ charcoal age, the onset of the Holocene

and the onset of *Corylus* pollen were set to 13,716, 11,557 and 10,675 a BP, respectively. However, in terms of deviation to the non-adjusted dates these are changes of +84, +93 and - 225 a. The adjustment of the onset of the *Corylus* pollen in Lake Gemündener Maar was therefore set to the maximum oldest age suggesting that for a more detailed driver analysis individual dated climate archives are urgently needed and that the results presented here should

not be over interpreted. Apart from these age-depth model uncertainties of the GM, we argue a conceivably driving mechanism for the $RH_{dv}$ variability in Central Europe could be a combination of solar activity changes, which triggered solar insolation, and North Atlantic Ocean temperature evolution. A key example might be the Preboreal Humid Phase. It can be expected that the North Atlantic Ocean, the main moisture source for Central Europe, revealed

already considerable higher temperatures during the Preboreal Humid Phase compared to the Younger Dryas, as indicated by a consistent ~ 2°C increase in planktonic foraminifera (*Globorotalia inflata*, *Globorotalia bulloides* and *Neogloboquadrina pachyderma*) derived Mg/Ca temperatures from a marine sediment core south of Iceland (Thornalley et al., 2009, 2010, 2011). In summary, this could lead to an enhanced moisture content of the atmosphere.

When these wetter air masses were transported on continental Europe, where low solar





insolation inhibited warming up and drying of these air masses, more humid climate conditions were established. Yet, it can be only speculated if the solar activity minimum during the early Younger Dryas (Fig. 6B) together with strong winds (Brauer et al., 2008) established climatic boundary conditions favoring sea water evaporation, which might have led to increased $RH_{dv}$

in Western Europe (Fig. 6A). Moreover, during the second phase of the Younger Dryas, in which a slight cooling of the North Atlantic Ocean coinciding with increasing solar activity, the climatic conditions on the European continent could have become drier, as suggested by the Lake Gemündener Maar $RH_{dv}$ record. In this light, the applied $\delta^2H_{n\text{-alkane}}$-$\delta^{18}O_{sugar}$ approach has great potential for establishing more quantitative relative humidity records, offering a

paleohygrometer approach for better understanding of past climate change.

## 4 Conclusions

Referring to the underlying research questions and based on the presented results and the outlined discussion (including the cited literature) the following conclusions have to be

drawn:

- The terrestrial vs. aquatic origin of bulk sedimentary organic matter cannot be determined unambiguously for the Gemündener Maar. This is caused by the bulk proxies (TOC/TN, $\delta^{13}C$ and $\delta^{15}N$) being not straightforward interpretable. At the same time, this also hampers inferring paleoclimate information based on those proxies. By

contrast, the alkane biomarkers with the chain-length $n\text{-C}_{27}$ and $n\text{-C}_{29}$ and the sugar biomarker arabinose can be most likely associated with the epicuticular leaf-wax layer and the hemicellulose structures of higher terrestrial plants, respectively. Therefore, they are assumed to originate primarily from leaf material of the Gemündener Maar catchment, which enabled us to use them in the coupled $\delta^2H$-$\delta^{18}O$ paleohygrometer

approach.

- Leaf water can be accessed from $\delta^2H_{n\text{-alkane}}$ ($n\text{-C}_{27}$ and $n\text{-C}_{29}$) and $\delta^{18}O_{sugar}$ (arabinose) by applying constant biosynthetic fractionation factors. We acknowledge that the assumption of constant fractionation factors introduces considerable uncertainties as highlighted by the broad literature discussion. Additionally, a correction for the

reconstructed leaf water isotope composition seems to be required to account for grassy $n$-alkane ($n\text{-C}_{27}$ and $n\text{-C}_{29}$) and arabinose contribution, because grasses are known to be less sensitive leaf water enrichment recorders.



- The reconstruction of relative humidity values is quite robust; it does not depend strongly on different scenarios used for calculating leaf water values. Furthermore, a correction to account for possible differences between Craig-Gordon modeled and measured leaf water isotope composition turned out to be dispensable for the here applied paleohyrometer approach. A constant leaf water evaporation line throughout the past seems to be a valid assumption. This was shown by alternative relative air humidity calculations based on apparent fractionation values, which account for a possible lack of equilibrium between plant source water and local atmospheric water vapour. The differences to the conventional reconstructed relative air humidity values (via Eq. 5) are far below the analytical errors associated with $\delta^2H_{n\text{-alkane}}$ and $\delta^{18}O_{sugar}$ measurements. Moreover, the used local meteoric water line of Germany seems to be a solid base line here. The reconstructed relative air humidity values are interpreted to reflect RH of the vegetation period and the daytime ($RH_{dv}$).

- The established Lake Gemündener Maar $RH_{dv}$ record supports a two-phasing of the Younger Dryas, i.e. a relative wet phase on Allerød level followed by a drier Younger Dryas ending. Overall drier climatic conditions throughout the Younger Dryas or a two-phasing in terms of a first dry and cold Younger Dryas phase followed by a warmer period along with increasing precipitation amounts could not be corroborated. Surprisingly, the amplitude of $RH_{dv}$ changes during the Early Holocene was more pronounced compared to the Younger Dryas. One possible driver for the unexpected Lake Gemündener Maar $RH_{dv}$ variations could be the solar activity, as derived from the comparison with IntCal [14]C production and Greenland ice-core [10]Be flux rate data.

**Data availability**

The data will be made available upon request.

**Author contributions**

J. Hepp and L. Wüthrich wrote the paper; M. Zech and R. Zech acquired financial support and wrote the paper. F. Sirocko was responsible for lake coring and provided the chronology and stratigraphy. L. Wüthrich, J. Hepp, T. Bromm, M. Bliedtner and I.K. Schäfer carried out laboratory work and did data evaluation. M. Bliedtner, I.K. Schäfer, B. Glaser, K. Rozanski, F. Sirocko contributed to the discussion of the data and commented on the manuscript.



## Competing interests

Competing financial interests: The authors declare no competing financial interests.

## Acknowledgements

We greatly acknowledge R. Muscheler (Lund University) for providing [10]Be flux as well as [14]C
production rates data. We thank M. Benesch (Martin-Luther-University Halle-Wittenberg), J.
Zech (University of Bern), S. Lutz (University of Zurich), N. Baltić and P. Kretschmer (Martin-
Luther-University Halle-Wittenberg) for their support during laboratory work. We thank P.M.
Grootes (Kiel University) for valuable recommendations during manuscript preparation and A.
Kühnel (Technical University of Munich) for helping us in establishing the Monte-Carlo-based
correlation scheme. Swiss National Science Foundation (PP00P2 150590) funds L. Wüthrich,
M. Bliedtner, I.K. Schäfer and R. Zech. Involvement of K. Rozanski was supported by AGH
UST statutory task No. 11.11.220.01/1 within subsidy of the Ministry of Science and Higher
Education. J. Hepp greatly acknowledges the support given by the German Federal
Environmental Foundation.

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



**Figures**

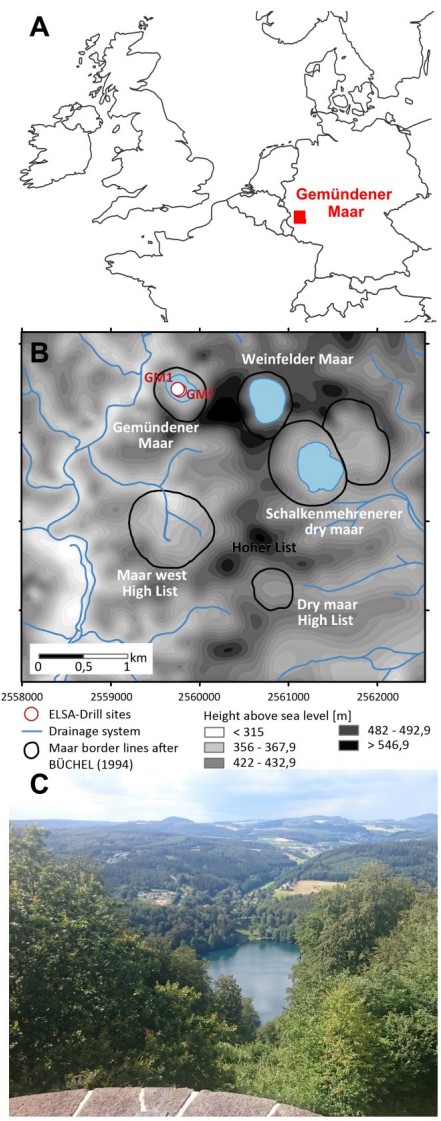

**Fig. 1**: (A) Location of the Lake Gemündener Maar in the Eifel Region in Germany (generated using OpenStreetMap homepage, www.openstreetmap.org). (B) Digital terrain model and
5    drainage system of the closer surrounding of Lake Gemündener Maar with maar borders according to Büchel (1994), representing the size of the crater. In addition, the core position is displayed (GM1; 50°10'39.853"N, 6°50'12.912"E) along with the short core GMf (not part of the study) marked as ELSA-Drill sites. Both cores are part of the Eifel Laminated Sediment Archive Project (ELSA-Project). (C) Photo of Lake Gemündener Maar showing the steep and
10    densely vegetated catchment (from M. Zech).





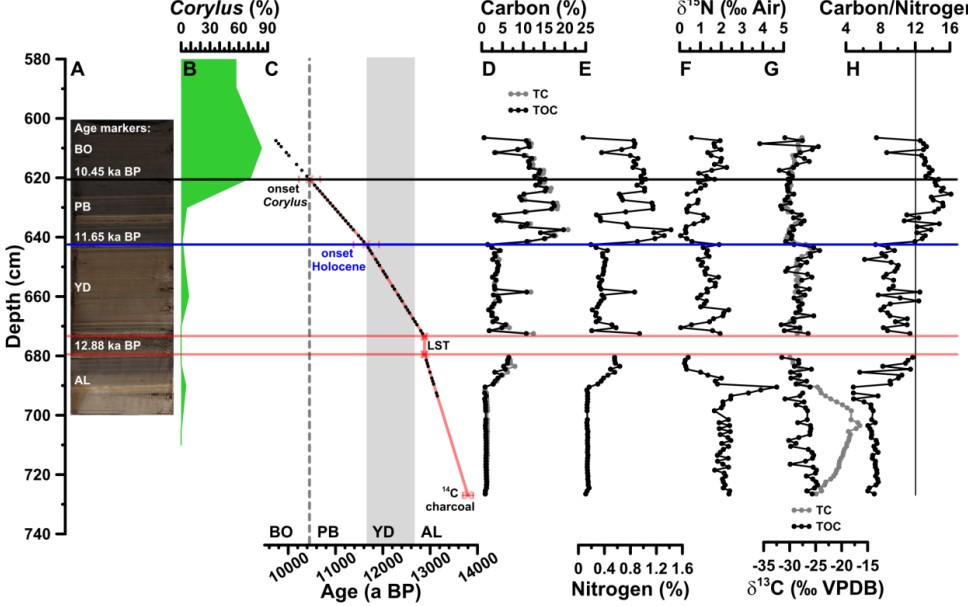

**Fig. 2**: (A) Photo of the investigated GM1 core section, with regard to the biomarkers (607 to 694 cm depth), including age markers. (B) Depth profile of *Corylus* pollen (derived from unpublished pollen analysis by F. Sirocko). (C) Age-depth model of the full investigated GM1 section (606 to 727 cm depth) consisting of a $^{14}$C dated piece of charcoal, the Laacher See Tephra (LST), the onset of the Holocene (blue line) and *Corylus* pollen (black line). Additionally, the biomarker sampling points (black points) are displayed. Depth profiles of (D) total and total organic carbon (TC, TOC), (E) total nitrogen (TN), (F) bulk stable nitrogen isotope composition ($\delta^{15}$N), (G) TC and TOC stable carbon isotope composition ($\delta^{13}$C$_{TC}$, $\delta^{13}$C$_{TOC}$), (H) Carbon to nitrogen ratio based on total organic carbon and total nitrogen (TOC/TN). The vertical line in (H) indicates a C/N ratio threshold of 12. AL = Allerød, YD = Younger Dryas, PB = Preboreal, and BO = Boreal.



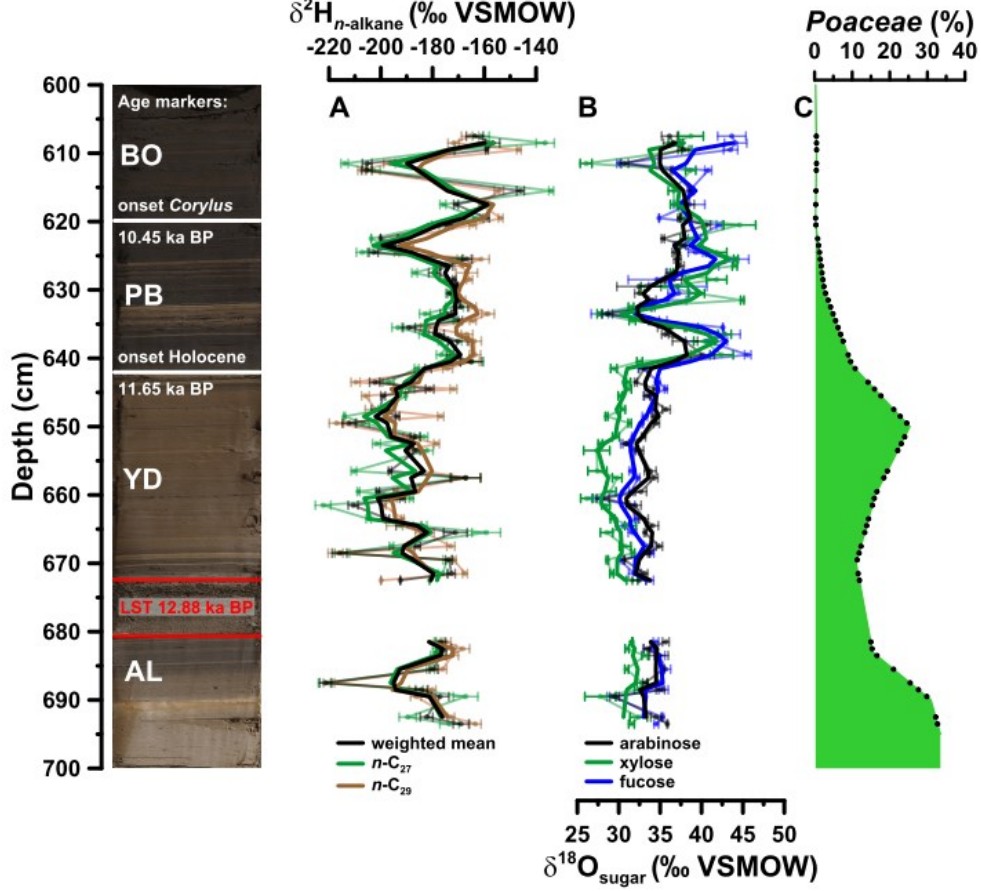

**Fig. 3**: (A) Depth profiles of compound-specific stable hydrogen isotope composition of the individual alkanes $n$-C$_{27}$ and $n$-C$_{29}$ and the weighted mean ($\delta^2$H$_{n\text{-alkane}}$). (B) Compound-specific stable oxygen composition of the individual sugars arabinose, xylose and fucose ($\delta^{18}$O$_{sugar}$). All plots are shown with error bars representing standard measurement errors overlain by three point moving averages (bold lines). (C) Depth profile of *Poaceae* pollen (derived from unpublished pollen analysis by F. Sirocko). Additionally, the resampled data points (black points) used for the grass correction procedures (Eq. 12 and 13) are displayed. In addition, the GM1 core picture with the used age markers is displayed. AL = Allerød, LST = Laacher See Tephra, YD = Younger Dryas, PB = Preboreal, and BO = Boreal.





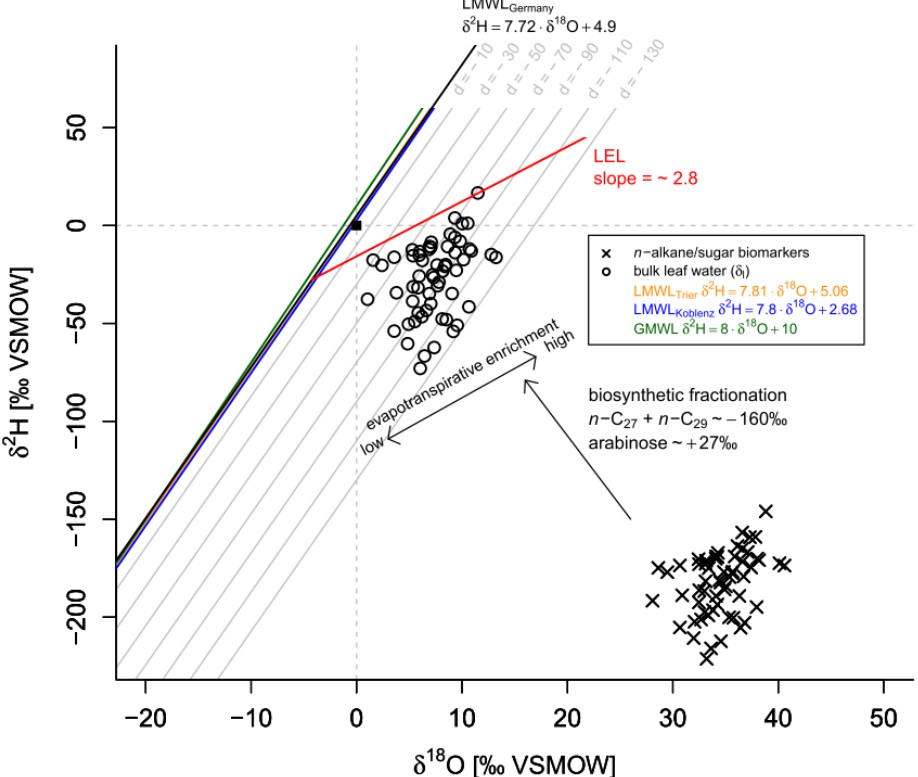

**Fig. 4**: (A) Conceptual framework of the applied $\delta^2H_{n\text{-alkane}}$-$\delta^{18}O_{sugar}$ approach displayed as $\delta^{18}O$-$\delta^2H$ diagram showing the measured $n$-alkanes (weighted mean of $n$-$C_{27}$ and $n$-$C_{29}$) and sugar (arabinose) biomarkers (black crosses), the reconstructed leaf water ($\delta_l$; open circles), the global meteoric water line (GMWL, green line) and the local meteoric water lines of Germany (LWML$_{Germany}$, black line), Trier (LWML$_{Trier}$, yellow line) and Koblenz (LWML$_{Koblenz}$, blue line). The arrows indicate natural processes of evaportranspirative enrichment of leaf water and biosynthetic fractionation during biomarker synthesis. Grey lines indicate the parallel distance between the individual reconstructed evaporative site leaf water points and the LMWL$_{Germany}$, expressed as d = $\delta^2H$ – 7.72 · $\delta^{18}O$, which can serve as proxy for mean daytime vegetation period relative humidity.



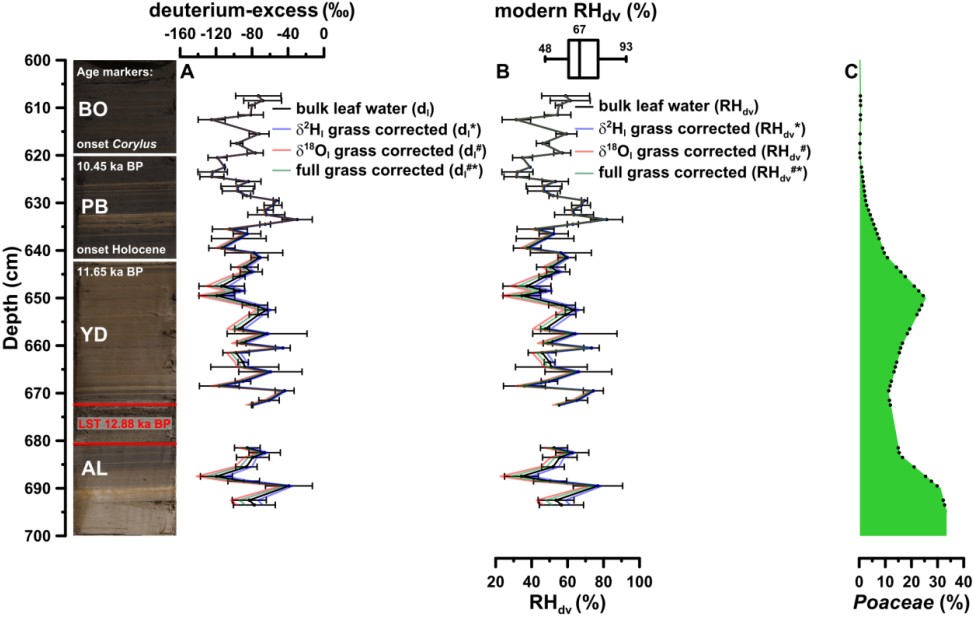

**Fig. 5**: (A) Reconstructed deuterium-excess depth profiles using leaf water ($d_l$; black line) in comparison to solely *n*-alkane grass corrected $d_l$* values (light blue line), solely arabinose grass corrected $d_l^{\#}$ values (light red line), and full grass corrected $d_l^{\#}$* values (light green line). The error bar of $d_l$ value are calculated according to Eq. (7). (B) The corresponding $RH_{dv}$ records to (A). Modern relative air humidity variability during daytime and vegetation period is displayed as boxplot derived from the adjacent meteorological station Nürburg-Barweiler, using monthly means from April to October between 6 a.m. and 7 p.m. (based on hourly data from 1995 to 2015; Deutscher Wetterdienst, 2016). The bold numbers within the boxplot represents the maximum, median, and minimum value, respectively. (C) Depth profile of *Poaceae* pollen (derived from unpublished pollen analysis by F. Sirocko). Additionally, the resampled data points (black points) are displayed. The GM1 core picture with the used age markers are displayed on the left. AL = Allerød, LST = Laacher See Tephra, YD = Younger Dryas, PB = Preboreal, and BO = Boreal.



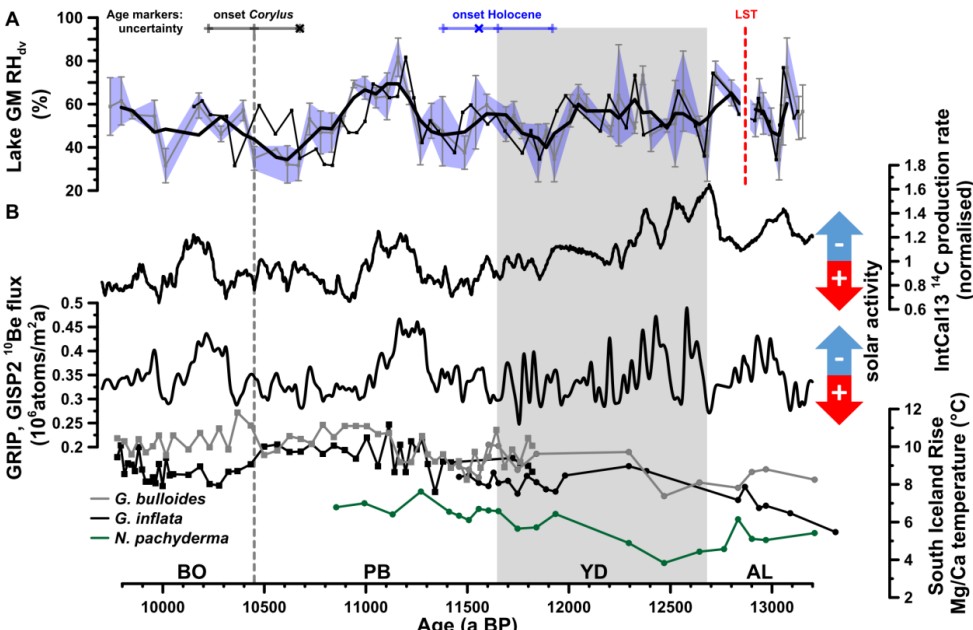

**Fig. 6**: (A) Reconstructed Lake Gemündener Maar (GM) $RH_{dv}$ values, overlain by a three point moving averages (thick black line). Error bars and the blue shaded area indicate uncertainties introduced by $^2H_{n\text{-alkane}}/^{18}O_{arabinose}$ measurements calculated according to error propagation (Eq.

7). In addition, the derived 68% uncertainty of the onset of *Corylus* pollen (black) and the Holocene (blue) are shown. The crosses within these lines represents the adjusted ages according to the Monte-Carlo-Simulation based correlation procedure to the IntCal13 $^{14}$C production rate. The GM $RH_{dv}$ values are additionally shown on an optimized age-depth scale (thin black line with squares). Please note that the $^{14}$C charcoal age is also covered by an error

and was as well included in the adjustment procedure but not shown in this graph for clarity reasons. (B) IntCal13 $^{14}$C production rate, Greenland ice-core (GRIP, GISP2) $^{10}$Be flux record (both from Muscheler et al., 2014) and South Iceland Rise planktic Mg/Ca derived water temperatures from RAPiD-12-1K (squares; 9,800 to 11,800 a BP) and RAPiD-15-4P (circles; 10,900 to 13,100 a BP). RAPiD-12-1K and RAPiD-15-4P *G. bulloides* and *G. inflat* data from

Thornalley et al. (2009) and Thornalley et al. (2010), respectively. RAPiD-15-4P *N. pachyderma* data from Thornalley et al. (2011). Note that each record is plotted on its own timescale (planktonic Mg/Ca data see (Thornalley et al., 2009, 2010), $^{10}$Be data on GICC05 (Rasmussen et al., 2006), $^{14}$C data on IntCal13 calibration curve (Reimer et al., 2013) and $RH_{dv}$ data on Lake GM age-depth model, see Fig. 2C). AL = Allerød, LST = Laacher See Tephra,

YD = Younger Dryas, PB = Preboreal, and BO = Boreal.





**Table**

**Tab. 1:** Scenarios (1-4) comprising the different reconstructed leaf water parameters based on the *n*-alkane/sugar biomarker, the respective equations used for this reconstruction, the resulting deuterium-excess of leaf water based on the reconstructed leaf water as input for Eq. (5) and the finally derived relative air humidity values during daytime and vegetation period ($RH_{dv}$) with this equation.

| scenario | leaf water reconstructed from *n*-alkane/sugar biomarkers | equations used for leaf water reconstruction | resulting deuterium-excess of leaf water as input for Eq. (5) | relative air humidity during daytime and vegetation period according Eq. (5) |
|---|---|---|---|---|
| 1 | $\delta^2 H_l / \delta^{18} O_l$ | (8) and (9) | $d_l$ | $RH_{dv}$ |
| 2 | $\delta^2 H_l * / \delta^{18} O_l$ | (8) and (9) + (11) | $d_l *$ | $RH_{dv} *$ |
| 3 | $\delta^2 H_l / \delta^{18} O_l {}^{\#}$ | (8) + (10) and (9) | $d_l {}^{\#}$ | $RH_{dv} {}^{\#}$ |
| 4 | $\delta^2 H_l * / \delta^{18} O_l {}^{\#}$ | (8) + (10) and (9) + (11) | $d_l *^{\#}$ | $RH_{dv} *^{\#}$ |