# Peer review of "Clim. Past Discuss., https://doi.org/10.5194/cp-2018-114 Manuscript under review for journal Clim. Past"

_Climate of the Past, 2018_

## Referee Comment (RC1) · Anonymous Referee #1 · 20 Sep 2018

General comments: The manuscript represents a substatial contribution to climate change, as it addresses the reconstruction of past humidity variations. So far, this climatic parameter could only be addressed based on pollen assemblages. The manuscript outlines the conceptual framework in great detail, and the results are appropriately discussed.

I have only one major concern: It is well known, that the dD of n-alkanes reflect source water isotopic composition (meteoric water in case of vegetation). The dD of rain water is related to Air Temperature, so the dD of source and related leaf water changes with

temperature. This would result in a shift of the intercept of LEL with the Meteoric Water Line towards more negative values during cold periods (Fig. 4 in the manuscript). However, the authors do not discuss the effect of this temperature dependency on the measured dD values of leaf-wax n-alkanes. At least it is not obvious from the discussion presented. In Fig. 3, it is shown that n-alkanes are depleted in D during Younger Dryas. For me, it is consistent with colder temperatures during this interval. The authors need to adress this topic, outlining if, and in what manner, temperature changes will affect the reconstructed humidity values in their approach.

Specific comments: The abbreviation LEL (Page 11, line 9) should be explained earlier (LEL = local evaporation line).

Technical comments: The manuscript is pretty long. The clearity of several points may be improved by shortening of the text. This may also facilitate a wider distribution of the paper.

---

## Short Comment (SC1) · 8 Oct 2018

The manuscript represents a major step forward with regard to the reconstruction of humidity variability and as such is of great interest for the scientific community. Aside from the isotopic part of the methods chapter being rather long (two chapters from page 6-12), the chronology has several weak points. On page 2, line 4-6 Gemündener Maar (GM) is mentioned in line with Holzmaar and Meerfelder Maar. These sites with natural eutrophication since the onset of the Lateglacial formed and preserved annually laminated sediments (ALS), which provided a precise time control for lacustrine systems.

This is completely different for GM, a lake that remained oligotrophic until World War II and only in the 1950ies was culturally eutrophicated, still without any deposition of ALS. This should be clarified in the text.

The chronology is a mixture of one radiocarbon date, the age of LST tephra, a TOC increase assumed for the onset of the Holocene and a pollen-inferred age for the PB/BO transition. Except for the age of the LST all other ages are questionable. 1. radiocarbon age: the age of 11,950 BP is from charcoal and as such potentially can be linked to reworking. One radiocarbon age to date the Lateglacial is not enough! And in general, why is there only one radiocarbon age for the entire record of ∼3500 years? At least during the early Holocene there should be enough terrestrial plant macrofossils for a proper dating. 2. How reliably is the TOC increase related to the onset of the PB? This is questionable but might easily be verified by pollen data. 3. The pollen age for the PB/BO transition (10,450 cal. BP) is in disagreement with published ages from Holzmaar (10,800 cal. BP) and Meerfelder Maar (10,740 cal. BP) - see Litt et al., 2009 (Boreas, 38: 679–690). Furthermore, the data to support this are not provided. And this is my major concern: in the manuscript reference is given to unpublished pollen data many times. Without pollen data being provided with the manuscript, not only the chronology remains unsupported by data.

Some minor issues: 1. On page 13, line 3ff the authors talk about clear evidence of carbonates. This is not at all evident from Fig. 2D. Moreover, where is the Ca coming from in this rather small catchment area composed of siliceous Devonian rocks? 2. The authors should explain, why they think that the record is representative for GM. Recovered from 20 m water depth (maximum water depth of GM is 39 m) in a small lake (diameter: 300 m), this implies a core from a relatively steep slope. 3. In Fig. 2H a threshold of 12 is used for C/N ratios related to autochtonous vs. allochthonous organic matter sources. The reference to Prahl et al. (1980) is related to estuarine sediments. Here the treshold of 10, e.g. Meyers (2003: Organic Geochemistry 34, 261– 289), should be preferred for lacustrine sediments.

---

## Referee Comment (RC2) · P.A. Meyers (Referee) · 9 Oct 2018

Hess and his colleagues present the elegant results of their compound-specific hydrogen and oxygen isotope analyses of 59 samples from a 87-cm interval of a sediment core from Lake Gemündener Maar in the Eifel region of Germany. The sediment in this core accumulated from the Allerød to the Boreal, and it appears to have been deposited continuously. The purpose of this study was to ascertain how dry the climate of the Younger Dryas may have been, and the isotopic compositions of the biomarker compounds were employed as paleohydrologic proxies. Two classes of biomarkers were

studied – n-alkanes derived from plant waxes and sugars derived from plant hemicellulose. The hydrogen isotopic compositions of the C27 and C29 n-alkanes and the oxygen isotopic compositions of fucose, xylose, and arabinose similarly shifted to lighter values in sediment samples from the Younger Dryas around the end of the Allerød, to return to heavier values at the beginning of the Preboreal. This pattern suggests that climate in the Eifel became less moist during the Younger Dryas. The isotopic changes are consistent with changes in pollen compositions that indicate local floral changes in response to climate changes.

Use of dual compound-specific isotopic analyses in paleoclimate reconstructions as done by Hess et al. is unusual and novel. Moreover, the authors have made very sophisticated interpretations of their data and have thoughtfully and carefully interpreted them. Unfortunately, their discussions are painfully detailed and make many sections nearly impenetrable. The manuscript is very hard to read and appreciate, and it does not have to be this way. Much of the detail is overkill – it could be trimmed or even omitted without weakening the authors' arguments and perhaps would strengthen their conclusions. Indeed, the conclusions are vague and not as clear as they should be.

I strongly suggest that the authors rewrite their very promising manuscript with the goal of cutting it by perhaps 30%. As one example, the long discussion of the source of the organic matter in the sediments is not necessary; all that is really needed is to establish that the biomarkers that have been analyzed are from vascular plants. As part of the rewriting, the English badly needs careful editing to smooth and refine it. Too many rough sections that detract from reading and appreciating the content of the text presently exist.

Phil Meyers

---

## Short Comment (SC2) · 15 Oct 2018

Hess and co-workers present a very interesting and novel dual compound-specific isotope approach to paleoclimate reconstructions. Through combining the hydrogen isotopic compositions of long-chain n-alkanes with the oxygen isotope composition of arabinose they were able to determine d-excess values for the YD – Holocene transition at lake Gemündener Maar and, subsequently, infer relative humidity (RH). Their reconstructed leaf water d-excess values range between -125 and -30 ‰. Given the novelty of the approach and the importance of the reconstructed d-excess values for

the following RH reconstruction an extensive discussion of the reliability of these values (irrespective of the analytical uncertainty) seems necessary and valuable. Is the wide range of reconstructed d-excess values for this site realistic, e.g. in view of modern leaf water d-excess? How can the large "short term" variations of d-excess be explained? How would the reconstructed d-excess values change if the biosynthetic fractionation values used would be different or impacted by vegetation change and thus lead to different sedimentary long-chain n-alkane ïĄďD values? Larger changes of the terrestrial and lacustrine vegetation in the course of the Lateglacial – Holocene transition are well known. Species-specific effects seem likely not only between grasses and woody plants (as discussed regarding an overall signal dampening), but also between different woody plants (Hou et al., 2007). In addition, in a lacustrine environment the long-chain n-alkane composition (n-C27,n-C29) of sedimentary organic matter might be influenced by aquatic macrophytes (e.g. Aichner et al., 2010). Since the magnitude of fractionation between source water and biochemical compounds seems to be distinctively different for terrestrial and aquatic plants (Chikaraishi et al., 2004) even small amounts of macrophyte derived long-chain n-alkanes might have a considerable impact on the total sedimentary long-chain n-alkane ïĄďD value. A sensitivity study could potentially identify the degree of uncertainty introduced by respective relevant factors. In the course of the YD- Holocene transition changes in precipitation isotopic composition have to be taken into account that will eventually translate into isotopic changes of source water (soil water, shallow groundwater) utilized by terrestrial vegetation. How would those changes impact on the reconstructed d-excess values? Direct comparison of station data with reconstructed RH seems difficult as the amplitude of station Rh data in monthly means is driven by the seasonal cycle whereas the reconstructed Rh values represent massively smoothed multiannual / multidecadal means (Fig. 5).

References Aichner, B., Herzschuh, U., Wilkes, H., Vieth, A. & Böhner, J. $\delta$D values of n-alkanes in Tibetan lake sediments and aquatic macrophytes - A surface sediment study and application to a 16 ka record from Lake Koucha. Organic Geochemistry

41, 779–790 (2010). Hou, J., D'Andrea, W.J., MacDonald, D., Huang, Y. Hydrogen isotopic variability in leaf waxes among terrestrial and aquatic plants around Blood Pond, Massachusetts (USA). Organic Geochemistry 38, 977-984 (2007). Chikaraishi, Y., Naraoka, H., Poulson, S.R. Hydrogen and carbon isotopic fractionations of lipid biosynthesis among terrestrial (C3, C4 and CAM) and aquatic plants. Phytochemistry 65, 1369–1381 (2004).

---

## Short Comment (SC3) · 25 Oct 2018

I disagree with the other comments that this is a novel approach. It is advocated by the authors since several years with various applications. My comment on it, however, remains the same each time: The approach cannot be done in soils or sediments as it compares apples with pears.

This is due to two reasons. First, plants incorporate the leaf water enrichment signal to variable degrees in their waxes and hemicellulose (Kahmen et al., 2013, Zech et

al., 2014). Leaf water is not the sole source of the hydrogen in waxes and oxygen in hemicellulose but a leaf water – xylem water mixture which is different between plants. It is not only grasses versus other plants as suggested here but various plants to a variable degree.

Second, plants produce waxes and hemicellulose in highly variable amounts (e.g. Diefendorf & Freimuth, 2017) depending on plant type and not correlated with each other, i.e. higher wax content is not necessarily associated to higher hemicellulose content.

In sedimentary archives or soils this means that the hydrogen isotope signal of leaf waxes is a wax-production weighted signal of the primary signal (temperature, amount, source effect) overprinted to a certain degree by evapo-transpiration and the hemicellulose oxygen isotope signal is a hemicellulose-production weighted signal of the same primary signal but affected to a different degree by evapo-transpiration due to different vegetation contributions to both parameters. Both $\delta$D of wax lipids and $\delta$18O of hemicellulose are thus qualitative hydrologic parameters that are not directly correlated and comparable. The position of the data points in $\delta$18O-$\delta$D space is thus dependent on vegetation composition and changes thereof and cannot be interpreted as reflecting leaf water isotopic enrichment in a quantitative approach. Application of such approach to sediments or soils will lead to erroneous and misleading interpretations.

Diefendorf, A.F., Freimuth, E.J., 2017. Extracting the most from terrestrial plant-derived n-alkyl lipids and their carbon isotopes from the sedimentary record: A review. Organic Geochemistry 103, 1-21.

Kahmen, A., Schefuß, E., Sachse, D., 2013. Leaf water deuterium enrichment shapes leaf wax n-alkane $\delta$D values of angiosperm plants I: Experimental evidence and mechanistic insights. Geochimica et Cosmochimica Acta 111, 39-49.

Zech, M., Mayr, C., Tuthorn, M., Leiber-Sauheitl, K., Glaser, B., 2014. Oxygen isotope ratios (18O/16O) of hemicellulose-derived sugar biomarkers in plants, soils and sediments as paleoclimate proxy I: Insight from a climate chamber experiment. Geochimica et Cosmochimica Acta 126, 614-623.

---

## Short Comment (SC4) · 31 Oct 2018

This is a joint comment by D. Sachse (GFZ Potsdam) and F. Schenk (Stockholm University - Bolin Center for Climate Research)

The authors are applying an approach to quantitively reconstruct relative humidity from terrestrial sedimentary archives based on biomarker $\delta$D and $\delta$18O values. In theory, this approach is elegant as it relies on two isotope systems and would represent a useful addition to our proxy portfolio, if it works. The underlying assumption is that

sedimentary n-alkanes and sugar biomarkers are equally sourced from the same vegetation, which is very unlikely to be actually the case. Earlier reviews and comments had already discussed some issues of this manuscript with regard to this assumption (see comment by E. Schefuß) as well as the chronological uncertainties due to the scarcity (or absence) of reliable age constraints (see comment by B. Zolitzschka), so we will not repeat those here.

In addition to those we would like to comment on some of the mechanistic data interpretations, which we think are not supported by the presented data and lack the context to earlier findings.

The authors argue their reconstructed relative humidity changes during the Late Glacial period and the early Holocene are driven by solar activity, which seems to be based on a perceived visual similarity of reconstructed 14C production rates and their relative humidity reconstruction.

In order to prove a mechanistic relationship between these parameters, two conditions have to be fulfilled:

1) objective demonstration of an actual covariation of these two parameters during the study period, e.g. a statistically significant correlation between the two parameters and 2) demonstration of a conceivable causal relation, i.e. a mechanism.

We think the current manuscript does not provide those, 1) is completely lacking and 2) is insufficient. We would recommend that a more detailed statistical analysis is presented than what is provided on page 27.

Page 27, lines 15-19: The way the Monte-Carlo-Simulations are conducted and the results should be explained in more detail here. The reported "maximum correlation coefficient of 0.37" does no tell much without providing the confidence limits of this test. If the test is based on smoothed values, the effective degrees of freedom may become very low and even r=0.4 may not be distinguishable from noise.

[Figure]

From visual inspection of Figure 6, there is no apparent consistent link between solar activity (IntCal13 14C production rate, Greenland 10Be flux) and RHdv. Based on the solar activity proxies, periods of negative activities are assumed for around 12.5 ka BP, 11.2 ka BP (roughly PB) and 10.2 ka BP (roughly BO). The low solar activity around the PB coincide with a period of very high RHdv while the low solar activity around 12.5 ka BP and 10.2 ka BP does not.

In addition, from Figure 6 it is not clear how Mg/Ca-based SST from South Iceland Rise should in any way be linked to RHdv. Neither the data show any convincing co-variability nor is there a plausible mechanism which should link these two regions. A more straightforward driver or humidity changes may be rather a southward migration of the North Atlantic sea-ice front in response to a weak AMOC state during the YD (e.g. Renssen et al. 2018). Overall, it is also odd to argue that "one possible driver for the unexpected Lake Gemündener Maar RHdv variations could be the solar activity" in the abstract without trying to explain why the same solar activity causes "expected" variations in other lakes including Lake Meerfelder Maar (which is a site closeby).

We note, that the chronological uncertainties due to the poorly constrained age model (see comment by B. Zolitzschka) make any correlation analysis difficult, so that 1) can possibly only be confirmed after the age model uncertainties are reduced.

Further, the authors suggest that their data show a wetter first and a dryer second phase of the Younger Dryas (YD) period, but provide no data on this, i.e. how much different those supposedly were. When looking at Fig. 6 it is difficult to actually see a change in reconstructed relative humidity (taking into account the error ranges) during the whole studied period, except for the apparent increase of values at around 11ka BP. If differences in a quantitative proxy are being discussed, these should be stated with actual values and uncertainties and be tested for actual statistically significant differences.

The interpretation of the derived relative humidity reconstruction, i.e. no change in

relative humidity at the YD onset, a wetter early YD and a dryer late YD and early Holocene, also should be discussed in the context of earlier findings. This interpretation disagrees with the bulk of previous literature data, based on palynological and geochemical evidence, which shows evidence for a dryer first half of the YD (in particular compared to the Allerød) and a wetter (and more variable) second half of the YD (Brauer et al., 1999; Bakke et al., 2009) in Europe. The authors mention this disagreement but provide no explanation (except that the other proxies are potentially biased). Also, new modelling results suggest an overshoot of humidity conditions at the Holocene / Younger Dryas boundary (Renssen et al., 2018) – a feature frequently captured in hydrological proxy data from ice cores (Rasmussen et al., 2006) to lacustrine sediments (Rach et al., 2017) but not apparently in the presented record.

If the interpretation of the data (see above) holds after statistical tests have been made, then these disagreements with existing proxy and modelling data need to be discussed. If no relationship can be statistically proven and no agreement with previous reconstructions is found, the potential of the proxy as a quantitative recorder of relative humidity should be re-evaluated.

Dirk Sachse & Frederik Schenk.

Bakke J., Lie Ø., Heegaard E., Dokken T., Haug G. H., Birks H. H., Dulski P. and Nilsen T. (2009) Rapid oceanic and atmospheric changes during the Younger Dryas cold period. Nat Geosci 2, 202–205.

Brauer A., Endres C., Gunter C., Litt T., Stebich M. and Negendank J. (1999) High resolution sediment and vegetation responses to Younger Dryas climate change in varved lake sediments from Meerfelder Maar, Germany. Quaternary Science Reviews 18, 321–329.

Rach O., Kahmen A., Brauer A. and Sachse D. (2017) A dual-biomarker approach for quantification of changes in relative humidity from sedimentary lipid D/H ratios. Clim. Past 13, 741–757.

Rasmussen S. O., Andersen K. K., Svensson A. M., Steffensen J. P., Vinther B. M., Clausen H. B., Siggaard-Andersen M.-L., Johnsen S. J., Larsen L. B., Dahl-Jensen D., Bigler M., Röthlisberger R., Fischer H., Goto-Azuma K., Hansson M. E. and Ruth U. (2006) A new Greenland ice core chronology for the last glacial termination. J. Geophys. Res. 111.

Renssen H., Goosse H., Roche D. M. and Seppä H. (2018) The global hydroclimate response during the Younger Dryas event. Quaternary Science Reviews 193, 84–97.

---

## Author Comment (AC1) · 5 Dec 2018

**Reply to Referee #1**

Johannes Hepp & Michael Zech & co-authors

*General comments: The manuscript represents a substatial contribution to climate change, as it addresses the reconstruction of past humidity variations. So far, this climatic parameter could only be addressed based on pollen assemblages. The manuscript outlines the conceptual framework in great detail, and the results are appropriately discussed.*

→ We are very grateful to anonymous Referee #1 for her/his constructive suggestions helping to improve our manuscript. Please find our replies to the individual comments below.

*I have only one major concern: It is well known, that the dD of n-alkanes reflect source water isotopic composition (meteoric water in case of vegetation). The dD of rain water is related to Air Temperature, so the dD of source and related leaf water changes with temperature. This would result in a shift of the intercept of LEL with the Meteoric Water Line towards more negative values during cold periods (Fig. 4 in the manuscript). However, the authors do not discuss the effect of this temperature dependency on the measured dD values of leaf-wax n-alkanes. At least it is not obvious from the discussion presented. In Fig. 3, it is shown that n-alkanes are depleted in D during Younger Dryas. For me, it is consistent with colder temperatures during this interval. The authors need to address this topic, outlining if, and in what manner, temperature changes will affect the reconstructed humidity values in their approach.*

→ We agree with Referee #1 that temperature is known to have a strong effect on $\delta^2H/\delta^{18}O_{precipitation}$ (and thus as well on $\delta^2H_{n\text{-alkane}}/\delta^{18}O_{sugar}$). Following her/his recommendation, we will readily include the reconstructed $\delta^2H/\delta^{18}O_{source\text{-water}}$ record for the Gemündener Maar as calculated from the intersects of the LELs with the GMWL/LMWL in the revised version. As it can be expected, that record shows generally more depleted values during the Younger Dryas, albeit the variability is clearly much higher than e.g. in the Greenland ice cores. Concerning the temperature effect on reconstructed humidity (RH) values in our approach, we explain in Chapter 3.3 that temperature changes of ±10 °C result in RH changes of just around ±1% by slightly

affecting the slope of the local evaporation line (LEL). We hope that this issue will become clearer during revision and shortening (as recommended by Referee #1) of our manuscript.

*Specific comments: The abbreviation LEL (Page 11, line 9) should be explained earlier (LEL = local evaporation line).*

→ Thanks. We will readily follow the recommendation of Referee #1 during revision.

*Technical comments: The manuscript is pretty long. The clearity of several points may be improved by shortening of the text. This may also facilitate a wider distribution of the paper.*

→ We agree with Referee #1. Following her/his recommendation, we will readily shorten the manuscript during revision and do our best in order to improve clarity.

---

## Author Comment (AC2) · 5 Dec 2018

**Reply to B. Zolitschka (SC1: Chronological issues)**

Johannes Hepp & Frank Sirocko & co-authors

*The manuscript represents a major step forward with regard to the reconstruction of humidity variability and as such is of great interest for the scientific community.*

→ Thank you B. Zolitschka for that constructive short comment. Please find our contribution to the discussion below. Please allow us to furthermore emphasize here, that the take home message of our manuscript, i.e. overall pronounced dry climatic conditions in Central Europe during the YD are not corroborated by our results, is not affected by the chronological uncertainties.

*Aside from the isotopic part of the methods chapter being rather long (two chapters from page 6-12), the chronology has several weak points. On page 2, line 4-6 Gemündener Maar (GM) is mentioned in line with Holzmaar and Meerfelder Maar. These sites with natural eutrophication since the onset of the Lateglacial formed and preserved annually laminated sediments (ALS), which provided a precise time control for lacustrine systems. This is completely different for GM, a lake that remained oligotrophic until World War II and only in the 1950ies was culturally eutrophicated, still without any deposition of ALS. This should be clarified in the text.*

→ The ELSA Project hast drilled a total of 17 sediment cores from 4 of the modern maar lakes (some of them presented in Sirocko et al., 2013, 2016), including Ulmener Maar, Schalkenmehrener Maar and Holzmaar. The ELSA core GM1 from Gemündener Maar is characterized by very well visible color and lithology changes, with all abrupt climate events and pollen zones of the early Holocene most clearly visible (Fig. 1 in the current manuscript). Accordingly, GM1 was chosen for this study. We will readily change the text in the revised manuscript in order to clearly differentiate between varve counted archives (e.g. Lake Holzmaar and Meerfelder Maar) and not varve counted ones (e.g. Lake Gemündener Maar, this study), as suggested.

*The chronology is a mixture of one radiocarbon date, the age of LST tephra, a TOC increase assumed for the onset of the Holocene and a pollen-inferred age for the PB/BO transition. Except*

*for the age of the LST all other ages are questionable. 1. radiocarbon age: the age of 11,950 BP is from charcoal and as such potentially can be linked to reworking. One radiocarbon age to date the Lateglacial is not enough! And in general, why is there only one radiocarbon age for the entire record of ~3500 years? At least during the early Holocene there should be enough terrestrial plant macrofossils for a proper dating. 2. How reliably is the TOC increase related to the onset of the PB? This is questionable but might easily be verified by pollen data. 3. The pollen age for the PB/BO transition (10,450 cal. BP) is in disagreement with published ages from Holzmaar (10,800 cal. BP) and Meerfelder Maar (10,740 cal. BP) - see Litt et al., 2009 (Boreas, 38: 679–690). Furthermore, the data to support this are not provided. And this is my major concern: in the manuscript reference is given to unpublished pollen data many times. Without pollen data being provided with the manuscript, not only the chronology remains unsupported by data.*

→ Indeed, the Lake Gemündener Maar core is not varve counted (despite it is varved/laminated to a very large extent). However, the Laacher See Tephra represents a perfect time marker, dated by Brauer et al. (1999) to 12,880 varve years BP in nearby Meerfelder Maar. The other time marker used to constrain the age model is the middle of the sharp increase in *Corylus* (hazel) pollen at 6.21 cm. In the revised version of the manuscript we will present a refined age-depth model based on new available pollen results from Lake Gemündener Maar, which provide now a higher resolution than the curves presented in the current manuscript. In the revised age-depth model we use now the sharp *Corylus* increase as time marker for the Preboreal/Boreal transition, which is indeed dated by Litt et al. (2009) to 10,740 varve years BP in the Lake Meerfelder Maar sediments. The offset of 60 years to the varve counted Holzmaar record of Zolitschka (1998), as it is presented by Litt et al. (2009), is within the uncertainty of placing the "onset" of the Preboreal in the Lake Gemündener Maar *Corylus* curve. A single radiocarbon age corroborates this stratigraphy in general, but is not used for the age model. Moreover, we will use the higher resolved pollen results to refine the begin of the Holocene (Younger Dryas/Preboreal transition) to 11,590 years BP, again with regard to Litt et al. (2009) and Zolitschka (1998), and no longer to 11,650 years BP according to Walker et al. (2009). It should be noted that the onset of the Holocene remained at the same depth as hitherto; and this nicely correlates with the TOC increase in the core, as before. Finally, we will readily provide the higher resolved pollen data/curves along with the refined age-depth model in the revised version of the manuscript, as suggested.

*Some minor issues:*

*1. On page 13, line 3ff the authors talk about clear evidence of carbonates. This is not at all evident from Fig. 2D. Moreover, where is the Ca coming from in this rather small catchment area composed of siliceous Devonian rocks?*

→ Please note that we argue in our manuscript, "$\delta^{13}C_{TC}$ clearly shows the presence of carbonate … (Fig. 2G)". We further state that this presence of carbonate is not well visible in the TC versus TOC record (Fig. 2D). However, this is just a scaling problem. For reasons of clarity, we will delete the TC curve in our revised manuscript. Ca was most likely primarily introduced into the Gemündener Maar by eolian processes (cf. the abundant literature about carbonate-containing loess in Central Europe).

*2. The authors should explain, why they think that the record is representative for GM. Recovered from 20 m water depth (maximum water depth of GM is 39 m) in a small lake (diameter: 300 m), this implies a core from a relatively steep slope.*

→ Sediment core GM1 is retrieved from a terrace on the steep slope of the maar at 20 m water depth, exactly in the fan of an underwater erosional gully. The sediments from the Laacher See tephra to the beginning of the Boreal have a sedimentation rate of ~0.33 mm/year. This sedimentation rate is lower than in the eutrophic varve counted lakes of Holzmaar and Meerfelder Maar, but the anoxia changes (causing the color changes) in the Gemündener Maar are very pronounced and occurred within a few years in this maar lake without any inlet and outlet. The Gemündener Maar sediments are, accordingly, not affected by fluvial sediment contributions.

*3. In Fig. 2H a threshold of 12 is used for C/N ratios related to autochtonous vs. allochthonous organic matter sources. The reference to Prahl et al. (1980) is related to estuarine sediments. Here the threshold of 10, e.g. Meyers (2003: Organic Geochemistry 34, 261– 289), should be preferred for lacustrine sediments.*

→ Thanks for that comment. We will readily change this during the revision of the result and discussion part of the manuscript.

→ Literature

Brauer, A., Endres, C., Günter, C., Litt, T., Stebich, M. and Negendank, J. F. W.: High resolution sediment and vegetation responses to Younger Dryas climate change in varved lake sediments from Meerfelder Maar, Germany, Quaternary Science Reviews, 18(3), 321–329, doi:10.1016/S0277-3791(98)00084-5, 1999.

Litt, T., Schölzel, C., Kühl, N. and Brauer, A.: Vegetation and climate history in the Westeifel Volcanic Field (Germany) during the past 11 000 years based on annually laminated lacustrine maar sediments, Boreas, 38(4), 679–690, doi:10.1111/j.1502-3885.2009.00096.x, 2009.

Sirocko, F., Dietrich, S., Veres, D., Grootes, P. M., Schaber-mohr, K., Seelos, K., Nadeau, M., Kromer, B., Rothacker, L., Röhner, M., Krbetschek, M., Appleby, P., Hambach, U., Rolf, C., Sudo, M. and Grim, S.: Multi-proxy dating of Holocene maar lakes and Pleistocene dry maar sediments in the Eifel , Germany, Quaternary Science Reviews, 62, 56–76, doi:10.1016/j.quascirev.2012.09.011, 2013.

Sirocko, F., Knapp, H., Dreher, F., Förster, M. W., Albert, J., Brunck, H., Veres, D., Dietrich, S., Zech, M., Hambach, U., Röhner, M., Rudert, S., Schwibus, K., Adams, C. and Sigl, P.: The ELSA-Vegetation-Stack: Reconstruction of Landscape Evolution Zones (LEZ) from laminated Eifel maar sediments of the last 60,000 years, Global and Planetary Change, 142, 108–135, doi:10.1016/j.gloplacha.2016.03.005, 2016.

Walker, M., Johnsen, S., Rasmussen, S. O., Popp, T., Steffensen, J.-P., Gibbard, P., Hoek, W., Lowe, J., Andrews, J., Björck, S., Cwynar, L. C., Hughen, K., Kershaw, P., Kromer, B., Litt, T., Lowe, D. J., Nakagawa, T., Newnham, R. and Schwander, J.: Formal definition and dating of the GSSP (Global Stratotype Section and Point) for the base of the Holocene using the Greenland NGRIP ice core, and selected auxiliary records, Journal of Quaternary Science, 24(1), 3–17, doi:10.1002/jqs, 2009.

Zolitschka, B.: A 14,000 year sediment yield record from western Germany based on annually laminated lake sediments, Geomorphology, 22(1), 1–17, doi:10.1016/S0169-555X(97)00051-2, 1998.

---

## Author Comment (AC3) · 5 Dec 2018

**Reply to P.A. Meyers (Referee #2)**

Johannes Hepp & Michael Zech & co-authors

*Hepp and his colleagues present the elegant results of their compound-specific hydrogen and oxygen isotope analyses of 59 samples from a 87-cm interval of a sediment core from Lake Gemündener Maar in the Eifel region of Germany. The sediment in this core accumulated from the Allerød to the Boreal, and it appears to have been deposited continuously. The purpose of this study was to ascertain how dry the climate of the Younger Dryas may have been, and the isotopic compositions of the biomarker compounds were employed as paleohydrologic proxies. Two classes of biomarkers were studied – n-alkanes derived from plant waxes and sugars derived from plant hemicellulose. The hydrogen isotopic compositions of the C27 and C29 n-alkanes and the oxygen isotopic compositions of fucose, xylose, and arabinose similarly shifted to lighter values in sediment samples from the Younger Dryas around the end of the Allerød, to return to heavier values at the beginning of the Preboreal. This pattern suggests that climate in the Eifel became less moist during the Younger Dryas. The isotopic changes are consistent with changes in pollen compositions that indicate local floral changes in response to climate changes.*

*Use of dual compound-specific isotopic analyses in paleoclimate reconstructions as done by Hepp et al. is unusual and novel. Moreover, the authors have made very sophisticated interpretations of their data and have thoughtfully and carefully interpreted them. Unfortunately, their discussions are painfully detailed and make many sections nearly impenetrable. The manuscript is very hard to read and appreciate, and it does not have to be this way. Much of the detail is overkill – it could be trimmed or even omitted without weakening the authors' arguments and perhaps would strengthen their conclusions. Indeed, the conclusions are vague and not as clear as they should be.*

*I strongly suggest that the authors rewrite their very promising manuscript with the goal of cutting it by perhaps 30%. As one example, the long discussion of the source of the organic matter in the sediments is not necessary; all that is really needed is to establish that the biomarkers that have been analyzed are from vascular plants. As part of the rewriting, the English badly needs careful editing to smooth and refine it. Too many rough sections that detract from reading and appreciating the content of the text presently exist.*

→ We are very grateful to P.A. Meyers (Referee #2) for his constructive and encouraging comments. We fully agree with him that the current version of our manuscript contains too many details, making it hard to read. Following his recommendation, we will readily shorten and rewrite our manuscript during revision.

---

## Author Comment (AC4) · 5 Dec 2018

**Reply to A. Lücke (SC2: d-excess reconstruction)**

Johannes Hepp & Michael Zech & co-authors

*Hepp and co-workers present a very interesting and novel dual compound-specific isotope approach to paleoclimate reconstructions. Through combining the hydrogen isotopic compositions of long-chain n-alkanes with the oxygen isotope composition of arabinose they were able to determine d-excess values for the YD – Holocene transition at lake Gemündener Maar and, subsequently, infer relative humidity (RH).*

→ We are very grateful to A. Lücke for his encouraging words and his constructive comments. Readily, we explain here in more detail why we consider our d-excess reconstruction to be robust and will include respective arguments in the revised manuscript where reasonable.

*Their reconstructed leaf water d-excess values range between -125 and -30‰. Given the novelty of the approach and the importance of the reconstructed d-excess values for the following RH reconstruction an extensive discussion of the reliability of these values (irrespective of the analytical uncertainty) seems necessary and valuable. Is the wide range of reconstructed d-excess values for this site realistic, e.g. in view of modern leaf water d-excess?*

→ Unfortunately, we are not aware of literature reporting on d-excess values of leaf water in or close-by to our study area. Still, our reconstructed d-excess values are realistic. For instance, Voelker et al. (2014) reported on 'deuterium deviations' (calculated as d-excess of leaf water minus 10‰). The deuterium deviation values in the respective publication (Voelker et al., 2014) range from ~0 to ~-200‰. Furthermore, Mayr (2002) conducted climate chamber experiments with *Vicia*, *Brassica* and *Eucalyptus* during his dissertation and measured $\delta^2H$ and $\delta^{18}O$ of leaf water ($\delta^{18}O$ of leaf water and of stem sugar biomarkers are published in Zech et al., 2014). Accordingly, d-excess of leaf water ranged from -38 to -171‰ and correlated highly significantly with RH (ranging from 21 to 68%). The former hence confirms that the reconstructed d-excess values of the Gemündener Maar ranging from -30 to -125‰ are well within the range that can be expected; the latter nicely corroborates the coupled $\delta^2H_{n\text{-alkane}}$-$\delta^{18}O_{sugar}$ paleohygrometer approach that is applied in

our manuscript in order to reconstruct RH values. A respective manuscript is currently under revision.

*How can the large "short term" variations of d-excess be explained?*

→ That's indeed a very interesting question. Concerning "short term", it should be noted that the resolution of our Gemündener Maar record is on average 58 years according to our age-depth model. Moreover, the reconstructed d-excess of leaf water reflects RH changes. In our opinion, RH changes can have two main reasons. First, moisture supply from the Atlantic Ocean is and was variable depending on sea surface temperature and on atmospheric circulation patterns. Second, we pointed in our manuscript to the finding that centennial variability of RH history of Western and Central Europe seems to covary with solar activity and thus insolation (Fig. 6). Possibly, insolation minima reduced the warming up and thus drying of air masses on the continent. Given that the solar activity records based on [10]Be and [14]C shown in Fig. 6 also reveal pronounced decadal variability, we suggest that decadal RH variability in our record may have been triggered by insolation changes.

*How would the reconstructed d-excess values change if the biosynthetic fractionation values used would be different or impacted by vegetation change and thus lead to different sedimentary long-chain n-alkane dD values? Larger changes of the terrestrial and lacustrine vegetation in the course of the Lateglacial – Holocene transition are well known. Species-specific effects seem likely not only between grasses and woody plants (as discussed regarding an overall signal dampening), but also between different woody plants (Hou et al., 2007). In addition, in a lacustrine environment the long-chain n-alkane composition (n-C27, n-C29) of sedimentary organic matter might be influenced by aquatic macrophytes (e.g. Aichner et al., 2010). Since the magnitude of fractionation between source water and biochemical compounds seems to be distinctively different for terrestrial and aquatic plants (Chikaraishi et al., 2004) even small amounts of macrophyte derived long-chain n-alkanes might have a considerable impact on the total sedimentary long-chain n-alkane dD value. A sensitivity study could potentially identify the degree of uncertainty introduced by respective relevant factors.*

→ We fully agree with A. Lücke that the biosynthetic fractionation factor used to reconstruct $\delta^2H_{leaf-water}$ values from $\delta^2H_{n-alkane}$ results (and thus d-excess) is one of the major uncertainties of the coupled $\delta^2H_{n-alkane}$-$\delta^{18}O_{sugar}$ paleohygrometer approach. In order to provide the magnitude: a shift in the $\delta^2H_{n-alkane}$ biosynthetic fractionation of ±10‰ corresponds to a change in the reconstructed RH value of ~±6%.

We acknowledge, that Hou et al. (2007) reported on significant variations in $\delta^2H_{n-alkane}$ within plant types. For instance, $\delta^2H$ of $n$-$C_{29}$ from trees ranged from -218 to -160‰ (mean = -185‰ ± 13‰). However, Hou et al. (2007) at the same time emphasized that the range of variation is clearly smaller than those among different plant types and that grasses have the lowest $\delta^2H$ values. $\delta^2H$ of $n$-$C_{29}$ in grasses ranged from -238 to -186‰ (mean = -219‰ ± 18‰). We therefore consider our calculations accounting for the dampening effect of leaf water enrichment (see Eqs. 10 and 11) as it occurs in grasses to be appropriate.

We are furthermore very grateful to A. Lücke for pointing to uncertainties associated with the biosynthetic fractionation factor for aquatic macrophytes. Usually, mid-chain $n$-alkanes such as $n$-$C_{23}$ are interpreted to be of aquatic origin, whereas the long-chain $n$-alkanes are interpreted to be of terrestrial origin. Given that (i) we found hardly any mid-chain $n$-alkanes in our archive and (ii) focus on the long-chain $n$-alkanes, we consider aquatic macrophytes to be of minor relevance in the case of our Gemündener Maar $\delta^2H_{n-alkane}$ record.

In our opinion, the issue raised by A. Lücke is much more problematic and may lead to erroneous interpretations when using $\delta^2H$ of $n$-$C_{23}$ for reconstructing $\delta^2H_{lake-water}$ (cf. "dual biomarker approach" of Rach et al., 2014, 2017), because Aichner et al. (2018) have recently shown for a lake in Poland that $n$-$C_{23}$ is a variable mixture of aquatic and terrestrial origin during the Late Glacial and Early Holocene. Namely, birch as pioneering and one of the dominant tree species during Late Glacial reforestation of Central Europe is known to produce considerable amounts of mid-chain $n$-alkanes (Tarasov et al., 2013). Albeit not included in the latter publication, $n$-$C_{23}$ concentrations of *Betula exilis* and *Betula pendula* reached 653 and even 2323 µg/g.

*In the course of the YD- Holocene transition changes in precipitation isotopic composition have to be taken into account that will eventually translate into isotopic changes of source water (soil*

*water, shallow groundwater) utilized by terrestrial vegetation. How would those changes impact on the reconstructed d-excess values?*

→ Yes, the YD-Holocene transition is indeed characterized by a shift towards more positive $\delta^2H/\delta^{18}O_{source-water}$ values as reconstructed by our coupled $\delta^2H_{n-alkane}$-$\delta^{18}O_{sugar}$ approach and as it can be expected. We will readily include this record in the revised manuscript. Importantly, however, the deuterium-excess variability of Greenland ice cores does not exceed 4‰ in the here relevant time period (see last paragraph of Chapter 3.3). Changes in $\delta^2H/\delta^{18}O_{precipitation/source\ water}$ have therefore a negligible effect on reconstructed d-excess and RH values.

*Direct comparison of station data with reconstructed RH seems difficult as the amplitude of station Rh data in monthly means is driven by the seasonal cycle whereas the reconstructed Rh values represent massively smoothed multiannual / multidecadal means (Fig. 5).*

→ Please note that our coupled $\delta^2H_{n-alkane}$ - $\delta^{18}O_{sugar}$ paleohygrometer approach reconstructs daytime RH values for the vegetation period (therefore abbreviated RH$_{dv}$) (cf. also Tuthorn et al., 2015). We therefore consider the comparison with station RH data, calculated using monthly means from April to October between 6 a.m. and 7 p.m. from 1995 to 2015, to be appropriate.

*References*

*Aichner, B., Herzschuh, U., Wilkes, H., Vieth, A. & Böhner, J. δD values of n-alkanes in Tibetan lake sediments and aquatic macrophytes - A surface sediment study and application to a 16 ka record from Lake Koucha. Organic Geochemistry 41, 779–790 (2010).*

*Hou, J., D'Andrea, W.J., MacDonald, D., Huang, Y. Hydrogen isotopic variability in leaf waxes among terrestrial and aquatic plants around Blood Pond, Massachusetts (USA). Organic Geochemistry 38, 977-984 (2007).*

*Chikaraishi, Y., Naraoka, H., Poulson, S.R. Hydrogen and carbon isotopic fractionations of lipid biosynthesis among terrestrial (C3, C4 and CAM) and aquatic plants. Phytochemistry 65, 1369–1381 (2004).*

→ Literature

Aichner, B., Ott, F., Słowiński, M., Noryśkiewicz, A. M., Brauer, A. and Sachse, D.: Leaf wax *n*-alkane distributions record ecological changes during the Younger Dryas at Trzechowskie paleolake (Northern Poland) without temporal delay, Climate of the Past Discussions, (March), 1–29, doi:10.5194/cp-2018-6, 2018.

Hou, J., D'Andrea, W. J., MacDonald, D. and Huang, Y.: Hydrogen isotopic variability in leaf waxes among terrestrial and aquatic plants around Blood Pond, Massachusetts (USA), Organic Geochemistry, 38(6), 977–984, doi:10.1016/j.orggeochem.2006.12.009, 2007.

Mayr, C.: Möglichkeiten der Klimarekonstruktion im Holozän mit $\delta^{13}$C- und $\delta^2$H-Werten von Baum-Jahrringen auf der Basis von Klimakammerversuchen und Rezentstudien, PhD thesis, Ludwig-Maximilians-Universität München. GSF-Bericht 14/02, 152 pp., 2002.

Rach, O., Brauer, A., Wilkes, H. and Sachse, D.: Delayed hydrological response to Greenland cooling at the onset of the Younger Dryas in western Europe, Nature Geoscience, 7(1), 109–112, doi:10.1038/ngeo2053, 2014.

Rach, O., Kahmen, A., Brauer, A. and Sachse, D.: A dual-biomarker approach for quantification of changes in relative humidity from sedimentary lipid D/H ratios, Climate of the Past, 13, 741–757, doi:10.5194/cp-2017-7, 2017.

Tarasov, P. E., Müller, S., Zech, M., Andreeva, D., Diekmann, B. and Leipe, C.: Last glacial vegetation reconstructions in the extreme-continental eastern Asia: Potentials of pollen and n-alkane biomarker analyses, Quaternary International, 290–291, 253–263, doi:10.1016/j.quaint.2012.04.007, 2013.

Voelker, S. L., Brooks, J. R., Meinzer, F. C., Roden, J., Pazdur, A., Pawelczyk, S., Hartsough, P., Snyder, K., Plavcova, L. and Santrucek, J.: Reconstructing relative humidity from plant $\delta^{18}$O and δD as deuterium deviations from the global meteoric water line, Ecological Applications, 24(5), 960–975, doi:10.1890/13-0988.1, 2014.

Zech, M., Mayr, C., Tuthorn, M., Leiber-Sauheitl, K. and Glaser, B.: Oxygen isotope ratios ($^{18}$O/$^{16}$O) of hemicellulose-derived sugar biomarkers in plants, soils and sediments as paleoclimate proxy I: Insight from a climate chamber experiment, Geochimica et Cosmochimica Acta, 126(0), 614–623, doi:http://dx.doi.org/10.1016/j.gca.2013.10.048, 2014.

---

## Author Comment (AC5) · 5 Dec 2018

**Reply to E. Schefuß (SC3: Vegetation effects on sedimentary organic isotope records)**

Johannes Hepp & Michael Zech & co-authors

*I disagree with the other comments that this is a novel approach. It is advocated by the authors since several years with various applications.*

→ We acknowledge that our coupled $\delta^2H_{n\text{-alkane}}$-$\delta^{18}O_{sugar}$ paleohygrometer approach was first published five years ago by Zech et al. (2013). A validation study using topsoils along a climate transect followed two years later by Tuthorn et al. (2015); in the same year, a modified version of the coupled approach was applied to lacustrine sediments (Hepp et al., 2015); and an application to a terrestrial paleosol sequence followed one year ago by Hepp et al. (2017).

Whether this approach is novel or not, should be seen in our opinion against the background that compound-specific $\delta^2H$ analysis using GC-Py-IRMS came up twenty years ago and meanwhile hundreds of papers dealing with $\delta^2H$ of leaf wax compounds or sedimentary hydrocarbons have been published. For recent reviews please see e.g. Pedentchouk and Zhou (2018) and Sessions (2016). In any case, we readily leave it to every reader to make his own opinion concerning the innovation of our approach.

*My comment on it, however, remains the same each time: The approach cannot be done in soils or sediments as it compares apples with pears. This is due to two reasons.*

→ Both *n*-alkane and sugar biomarkers that are produced in leaves reflect the isotopic composition of precipitation/plant source water modified by (primarily RH-dependent) evapotranspirative enrichment of leaf water. Therefore, it is difficult for us to follow/understand the argumentation of E. Schefuß concerning apples and pears.

→ We furthermore disagree with E. Schefuß that our coupled $\delta^2H_{n\text{-alkane}}$-$\delta^{18}O_{sugar}$ paleohygrometer approach cannot be applied to soils and sediments. Please allow us to refer once again to the validation paper of Tuthorn et al. (2015). In that study, *n*-alkane and sugar biomarkers were extracted from topsoils along a climate gradient in S-America

covering different vegetation types. Reconstructed RH values based on our coupled $\delta^2 H_{n\text{-}alkane}$-$\delta^{18}O_{sugar}$ paleohygrometer approach correlated highly significantly with actual RH values (R = 0.79, p < 0.001, n = 20).

*First, plants incorporate the leaf water enrichment signal to variable degrees in their waxes and hemicellulose (Kahmen et al., 2013, Zech et al., 2014). Leaf water is not the sole source of the hydrogen in waxes and oxygen in hemicellulose but a leaf water – xylem water mixture which is different between plants. It is not only grasses versus other plants as suggested here but various plants to a variable degree.*

→ This comment and statement are surprising and puzzling to us, because E. Schefuß is co-author of Kahmen et al. (2013). In that publication, the abstract reads "For dicotyledonous plants we found that the full extent of leaf water evaporative D-enrichment is recorded in leaf wax *n*-alkane δD values. For monocotyledonous plants [such as grasses], we found that between 18% and 68% of the D-enrichment in leaf water was recorded in the δD values of their *n*-alkanes." Concerning Zech et al. (2014), that paper dealt with stem material not leaf material. While the former does show a dampening effect, the latter doesn't. Hence, neither evidence nor literature are provided by E. Schefuß supporting his statement that other plants than grasses (for which we included correction calculations in our manuscript) incorporate noteworthy amounts of the xylem water signal in their leaf biomarkers.

*Second, plants produce waxes and hemicellulose in highly variable amounts (e.g. Diefendorf & Freimuth, 2017) depending on plant type and not correlated with each other, i.e. higher wax content is not necessarily associated to higher hemicellulose content. In sedimentary archives or soils this means that the hydrogen isotope signal of leaf waxes is a wax-production weighted signal of the primary signal (temperature, amount, source effect) overprinted to a certain degree by evapo-transpiration and the hemicellulose oxygen isotope signal is a hemicellulose-production weighted signal of the same primary signal but affected to a different degree by evapo-transpiration due to different vegetation contributions to both parameters. Both δD of wax lipids and δ18O of hemicellulose are thus qualitative hydrologic parameters that are not directly correlated and comparable. The position of the data points in δ18O-δD space is thus dependent on vegetation composition and changes thereof and cannot be interpreted as reflecting leaf water*

*isotopic enrichment in a quantitative approach. Application of such approach to sediments or soils will lead to erroneous and misleading interpretations.*

→ As long as only dicotyledonous plants are investigated or their leaves contributed primarily to a sedimentary archive, E. Schefuß is wrong in his statement that the data points in a $\delta^2$H - $\delta^{18}$O diagram are noteworthy affected by vegetation changes. Why should a variable production of *n*-alkane or sugar biomarkers affect the $\delta^2$H/$\delta^{18}$O values? E. Schefuß does not provide any evidence or literature supporting this statement.

→ The issue raised by E. Schefuß would become only relevant when grasses/monocotyledonous plants and at the same time coniferous trees are the primary sources of biomarkers to a sedimentary archive/soil (this does not apply to the Gemündener Maar according to the pollen results indicating strong presence of *Betula* during the Late Glacial). This assessment is based on the notion that except for *Juniperus*, conifers produce very low amounts of *n*-alkanes (e.g. Zech et al., 2012). In such cases, sugars will show a mixed $\delta^{18}$O signal of conifer needles and grasses (and thus partly a dampened leaf water enrichment signal), whereas *n*-alkanes will show the dampened $\delta^2$H signal of the grasses. As a result, reconstructed RH values will underestimate actual RH values. This explanation is corroborated by data obtained for topsoils along a European climate transect; the respective manuscript will be submitted during the next weeks.

*Diefendorf, A.F., Freimuth, E.J., 2017. Extracting the most from terrestrial plant-derived n-alkyl lipids and their carbon isotopes from the sedimentary record: A review. Organic Geochemistry 103, 1-21.*

*Kahmen, A., Schefuß, E., Sachse, D., 2013. Leaf water deuterium enrichment shapes leaf wax n-alkane _D values of angiosperm plants I: Experimental evidence and mechanistic insights. Geochimica et Cosmochimica Acta 111, 39-49.*

*Zech, M., Mayr, C., Tuthorn, M., Leiber-Sauheitl, K., Glaser, B., 2014. Oxygen isotope ratios (18O/16O) of hemicellulose-derived sugar biomarkers in plants, soils and sediments as paleoclimate proxy I: Insight from a climate chamber experiment. Geochimica et Cosmochimica Acta 126, 614-623.*

→ References

Hepp, J., Tuthorn, M., Zech, R., Mügler, I., Schlütz, F., Zech, W. and Zech, M.: Reconstructing lake evaporation history and the isotopic composition of precipitation by a coupled $\delta^{18}O–\delta^2H$ biomarker approach, Journal of Hydrology, 529, 622–631, 2015.

Hepp, J., Zech, R., Rozanski, K., Tuthorn, M., Glaser, B., Greule, M., Keppler, F., Huang, Y., Zech, W. and Zech, M.: Late Quaternary relative humidity changes from Mt. Kilimanjaro, based on a coupled $^2H$-$^{18}O$ biomarker paleohygrometer approach, Quaternary International, 438, 116–130, doi:10.1016/j.quaint.2017.03.059, 2017.

Kahmen, A., Schefuß, E. and Sachse, D.: Leaf water deuterium enrichment shapes leaf wax *n*-alkane δD values of angiosperm plants I: Experimental evidence and mechanistic insights, Geochimica et Cosmochimica Acta, 111, 39–49, doi:10.1016/j.gca.2012.09.004, 2013.

Pedentchouk, N. and Zhou, Y.: Factors Controlling Carbon and Hydrogen Isotope Fractionation During Biosynthesis of Lipids by Phototrophic Organisms, in Hydrocarbons, Oils and Lipids: Diversity, Origin, Chemistry and Fate. Handbook of Hydrocarbon and Lipid Microbiology, edited by H. Wilkes, pp. 1–24, Springer, Cham., 2018.

Sessions, A. L.: Factors controlling the deuterium contents of sedimentary hydrocarbons, Organic Geochemistry, 96(March), 43–64, doi:10.1016/j.orggeochem.2016.02.012, 2016.

Tuthorn, M., Zech, R., Ruppenthal, M., Oelmann, Y., Kahmen, A., del Valle, H. F., Eglinton, T., Rozanski, K. and Zech, M.: Coupling $\delta^2H$ and $\delta^{18}O$ biomarker results yields information on relative humidity and isotopic composition of precipitation - a climate transect validation study, Biogeosciences, 12, 3913–3924, doi:10.5194/bg-12-3913-2015, 2015.

Zech, M., Rass, S., Buggle, B., Löscher, M. and Zöller, L.: Reconstruction of the late Quaternary paleoenvironments of the Nussloch loess paleosol sequence, Germany, using n-alkane biomarkers, Quaternary Research, 78(2), 226–235,

doi:10.1016/j.yqres.2012.05.006, 2012.

Zech, M., Tuthorn, M., Detsch, F., Rozanski, K., Zech, R., Zöller, L., Zech, W. and Glaser, B.: A 220 ka terrestrial $\delta^{18}O$ and deuterium excess biomarker record from an eolian permafrost paleosol sequence, NE-Siberia, Chemical Geology, doi:10.1016/j.chemgeo.2013.10.023, 2013.

Zech, M., Mayr, C., Tuthorn, M., Leiber-Sauheitl, K. and Glaser, B.: Oxygen isotope ratios ($^{18}O/^{16}O$) of hemicellulose-derived sugar biomarkers in plants, soils and sediments as paleoclimate proxy I: Insight from a climate chamber experiment, Geochimica et Cosmochimica Acta, 126(0), 614–623, doi:http://dx.doi.org/10.1016/j.gca.2013.10.048, 2014.

---

## Author Comment (AC6) · 5 Dec 2018

**Reply to D. Sachse and F. Schenk (SC4: Data analysis and paleoclimatic context)**

*The authors are applying an approach to quantitively reconstruct relative humidity from terrestrial sedimentary archives based on biomarker δD and δ18O values. In theory, this approach is elegant as it relies on two isotope systems and would represent a useful addition to our proxy portfolio, if it works. The underlying assumption is that sedimentary n-alkanes and sugar biomarkers are equally sourced from the same vegetation, which is very unlikely to be actually the case. Earlier reviews and comments had already discussed some issues of this manuscript with regard to this assumption (see comment by E. Schefuß) as well as the chronological uncertainties due to the scarcity (or absence) of reliable age constraints (see comment by B. Zolitzschka), so we will not repeat those here.*

→ We thank D. Sachse and F. Schenk for their comment and the possibility to emphasize here again, that our coupled $δ^2H_{n\text{-alkane}}$-$δ^{18}O_{sugar}$ paleohygrometer approach does work. The approach is based on our current knowledge about (i) RH-dependent leaf water enrichment and (ii) the incorporation of the $δ^2H/δ^{18}O_{leaf\ water}$ signal into leaf-derived *n*-alkane and sugar biomarkers. Uncertainties and systematic offsets need, of course, to be considered (see e.g. our reply to the comment of E. Schefuß concerning grasses). We agree with the Referees #1 and #2, that the current manuscript suffers from a too detailed description and discussion of methodological issues and will therefore follow their recommendation to shorten our manuscript. Most methodological details are anyway already published. Please allow us to point here again to the validation study of Tuthorn et al. (2015). In that study, *n*-alkane and sugar biomarkers were extracted from topsoils along a climate gradient in S-America covering different vegetation types. Reconstructed RH values based on our coupled $δ^2H_{n\text{-alkane}}$-$δ^{18}O_{sugar}$ paleohygrometer approach correlated highly significantly with actual RH values (R = 0.79, p < 0.001, n = 20).
Concerning the chronological uncertainties, please note that they do not limit the take home message of our manuscript, i.e. overall pronounced dry climatic conditions during the YD are not corroborated by our results (see also our reply to the comment of B. Zoliztschka).

*In addition to those we would like to comment on some of the mechanistic data interpretations, which we think are not supported by the presented data and lack the context to earlier findings.*

*The authors argue their reconstructed relative humidity changes during the Late Glacial period and the early Holocene are driven by solar activity, which seems to be based on a perceived visual similarity of reconstructed 14C production rates and their relative humidity reconstruction. In order to prove a mechanistic relationship between these parameters, two conditions have to be fulfilled:*

*1) objective demonstration of an actual covariation of these two parameters during the study period, e.g. a statistically significant correlation between the two parameters and 2) demonstration of a conceivable causal relation, i.e. a mechanism.*

*We think the current manuscript does not provide those, 1) is completely lacking and 2) is insufficient. We would recommend that a more detailed statistical analysis is presented than what is provided on page 27.*

*Page 27, lines 15-19: The way the Monte-Carlo-Simulations are conducted and the results should be explained in more detail here. The reported "maximum correlation coefficient of 0.37" does no tell much without providing the confidence limits of this test. If the test is based on smoothed values, the effective degrees of freedom may become very low and even r=0.4 may not be distinguishable from noise.*

*From visual inspection of Figure 6, there is no apparent consistent link between solar activity (IntCal13 14C production rate, Greenland 10Be flux) and RHdv. Based on the solar activity proxies, periods of negative activities are assumed for around 12.5 ka BP, 11.2 ka BP (roughly PB) and 10.2 ka BP (roughly BO). The low solar activity around the PB coincide with a period of very high RHdv while the low solar activity around 12.5 ka BP and 10.2 ka BP does not.*

*In addition, from Figure 6 it is not clear how Mg/Ca-based SST from South Iceland Rise should in any way be linked to RHdv. Neither the data show any convincing covariability nor is there a plausible mechanism which should link these two regions. A more straightforward driver or humidity changes may be rather a southward migration of the North Atlantic sea-ice front in response to a weak AMOC state during the YD (e.g. Renssen et al. 2018). Overall, it is also odd to argue that "one possible driver for the unexpected Lake Gemündener Maar RHdv variations could be the solar activity" in the abstract without trying to explain why the same solar activity causes "expected" variations in other lakes including Lake Meerfelder Maar (which is a site closeby).*

*We note, that the chronological uncertainties due to the poorly constrained age model (see comment by B. Zolitzschka) make any correlation analysis difficult, so that 1) can possibly only be confirmed after the age model uncertainties are reduced.*

*Further, the authors suggest that their data show a wetter first and a dryer second phase of the Younger Dryas (YD) period, but provide no data on this, i.e. how much different those supposedly were. When looking at Fig. 6 it is difficult to actually see a change in reconstructed relative humidity (taking into account the error ranges) during the whole studied period, except for the apparent increase of values at around 11ka BP. If differences in a quantitative proxy are being discussed, these should be stated with actual values and uncertainties and be tested for actual statistically significant differences.*

*The interpretation of the derived relative humidity reconstruction, i.e. no change in relative humidity at the YD onset, a wetter early YD and a dryer late YD and early Holocene, also should be discussed in the context of earlier findings. This interpretation disagrees with the bulk of previous literature data, based on palynological and geochemical evidence, which shows evidence for a dryer first half of the YD (in particular compared to the Allerød) and a wetter (and more variable) second half of the YD (Brauer et al., 1999; Bakke et al., 2009) in Europe. The authors mention this disagreement but provide no explanation (except that the other proxies are potentially biased). Also, new modelling results suggest an overshoot of humidity conditions at the Holocene / Younger Dryas boundary (Renssen et al., 2018) – a feature frequently captured in hydrological proxy data from ice cores (Rasmussen et al., 2006) to lacustrine sediments (Rach et al., 2017) but not apparently in the presented record.*

*If the interpretation of the data (see above) holds after statistical tests have been made, then these disagreements with existing proxy and modelling data need to be discussed. If no relationship can be statistically proven and no agreement with previous reconstructions is found, the potential of the proxy as a quantitative recorder of relative humidity should be re-evaluated.*

*Dirk Sachse & Frederik Schenk.*

*Bakke J., Lie Ø., Heegaard E., Dokken T., Haug G. H., Birks H. H., Dulski P. and Nilsen T. (2009) Rapid oceanic and atmospheric changes during the Younger Dryas cold period. Nat Geosci 2, 202–205.*

*Brauer A., Endres C., Gunter C., Litt T., Stebich M. and Negendank J. (1999) High resolution sediment and vegetation responses to Younger Dryas climate change in varved lake sediments from Meerfelder Maar, Germany. Quaternary Science Reviews 18, 321–329.*

*Rach O., Kahmen A., Brauer A. and Sachse D. (2017) A dual-biomarker approach for quantification of changes in relative humidity from sedimentary lipid D/H ratios. Clim. Past 13, 741–757.*

*Rasmussen S. O., Andersen K. K., Svensson A. M., Steffensen J. P., Vinther B. M., Clausen H. B., Siggaard-Andersen M.-L., Johnsen S. J., Larsen L. B., Dahl-Jense D., Bigler M., Röthlisberger R., Fischer H., Goto-Azuma K., Hansson M. E. and Ruth U. (2006) A new Greenland ice core chronology for the last glacial termination. J. Geophys. Res. 111.*

*Renssen H., Goosse H., Roche D. M. and Seppä H. (2018) The global hydroclimate response during the Younger Dryas event. Quaternary Science Reviews 193, 84–97.*

→ Statistical data analysis: Following the recommendation of the Referees #1 and #2 we will shorten our manuscript during revision. The Monte-Carlo-Simulation-based correlation analysis will not be included any longer. Still, we are convinced that we should point our readers to the resemblance of the Gemündener Maar RH record with the solar activity records of Rasmussen et al. (2006) and Reimer et al. (2013), as presented by Muscheler et al. (2014).

→ Mechanism for solar activity/insolation effect on RH: Please see page 27, lines 28ff, where we explain that "It can be expected that the North Atlantic Ocean, the main moisture source for Central Europe, revealed already considerable higher temperatures during the Preboreal Humid Phase compared to the Younger Dryas, as indicated by a consistent ~ 2°C increase in planktonic foraminifera (*Globorotalia inflata*, *Globorotalia bulloides* and *Neogloboquadrina pachyderma*) derived Mg/Ca temperatures from a marine sediment core south of Iceland (Thornalley et al., 2009, 2010, 2011). […] This could lead to an enhanced moisture content of the atmosphere. When these wet air masses were transported to continental Europe, where low solar insolation inhibited warming up and drying of these air masses, more humid climate conditions were established." Please also note that the main focus of our manuscript is to establish an RH record for Central Europe, not paleoclimate modeling. Still, we would be very delighted, of course, if experts in the

field of paleoclimate modeling such as Renssen and co-workers can include and possibly evaluate our findings in their models.

→ Paleoclimatic context / earlier findings and proxy evaluation:

First, please allow us to point again to evidence from pollen and biomarker results supporting a first wet Younger Dryas followed by a more drier Younger Dryas ending (see the section 3.4 of the current manuscript, which includes the references Isarin et al., 1998; Muschitiello et al., 2015).

Second, we highly welcome a critical evaluation of our proxy record as well as of all other proxy records established for the YD. For instance, the disagreement of our RH record with the one established by Rach et al. (2014, 2017) for the neighboring Meerfelder Maar might be explained with the latter authors using $n$-$C_{23}$ as aquatic-derived $n$-alkane in order to reconstruct $\delta^2 H_{lake\text{-}water}$. However, there is increasing evidence that $n$-$C_{23}$ is also of terrestrial origin. Aichner et al. (2018) have recently shown for a lake in Poland that $n$-$C_{23}$ is a variable mixture of aquatic and terrestrial origin in those Late Glacial and Early Holocene sediments. And birch as pioneering and one of the dominant tree species during Late Glacial reforestation of Central Europe is known to produce considerable amounts of mid-chain $n$-alkanes (Tarasov et al., 2013). Albeit not included in the latter publication, $n$-$C_{23}$ concentrations of *Betula exilis* and *Betula pendula* reached 653 and even 2323 μg/g in that study. This is highly relevant, because as emphasized by A. Lücke in his comment, aquatic and terrestrial $n$-alkanes have different biosynthetic fractionation factors. Small changes in the contribution of terrestrial vs. aquatic $n$-alkanes will thus have a considerable impact on the reconstructed $\delta^2 H$ ($n$-$C_{23}$) record.

Anyway, we wish both the 'dual biomarker approach' of Rach et al. (2014, 2017) and our 'coupled $\delta^2 H_{n\text{-}alkane}$-$\delta^{18} O_{sugar}$ paleohygrometer approach' to be further applied and tested and are very much looking forward to learn how the discrepancy in the current state of research concerning RH history during the YD will be solved in the future.

Johannes Hepp & Michael Zech & co-authors

→ References

Aichner, B., Ott, F., Słowiński, M., Noryśkiewicz, A. M., Brauer, A. and Sachse, D.: Leaf

wax *n*-alkane distributions record ecological changes during the Younger Dryas at Trzechowskie paleolake (Northern Poland) without temporal delay, Climate of the Past Discussions, (March), 1–29, doi:10.5194/cp-2018-6, 2018.

Isarin, R. F. B., Renssen, H. and Vandenberghe, J.: The impact of the North Atlantic Ocean on the Younger Dryas climate in northwestern and central Europe, Journal of Quaternary Science, 13(5), 447–453, doi:10.1002/(sici)1099-1417(1998090)13:5<447::aid-jqs402>3.0.co;2-b, 1998.

Muscheler, R., Adolphi, F. and Knudsen, M. F.: Assessing the differences between the IntCal and Greenland ice-core time scales for the last 14,000 years via the common cosmogenic radionuclide variations, Quaternary Science Reviews, 106, 81–87, doi:10.1016/j.quascirev.2014.08.017, 2014.

Muschitiello, F., Pausata, F. S. R., Watson, J. E., Smittenberg, R. H., Salih, A. A. M., Brooks, S. J., Whitehouse, N. J., Karlatou-Charalampopoulou, A. and Wohlfarth, B.: Fennoscandian freshwater control on Greenland hydroclimate shifts at the onset of the Younger Dryas, Nature Communications, 6, 8939, doi:10.1038/ncomms9939, 2015.

Rach, O., Brauer, A., Wilkes, H. and Sachse, D.: Delayed hydrological response to Greenland cooling at the onset of the Younger Dryas in western Europe, Nature Geoscience, 7(1), 109–112, doi:10.1038/ngeo2053, 2014.

Rach, O., Kahmen, A., Brauer, A. and Sachse, D.: A dual-biomarker approach for quantification of changes in relative humidity from sedimentary lipid D/H ratios, Climate of the Past, 13, 741–757, doi:10.5194/cp-2017-7, 2017.

Rasmussen, S. O., Andersen, K. K., Svensson, A. M., Steffensen, J. P., Vinther, B. M., Clausen, H. B., Siggaard-Andersen, M.-L., Johnsen, S. J., Larsen, L. B., Dahl-Jensen, D., Bigler, M., Röthlisberger, R., Fischer, H., Goto-Azuma, K., Hansson, M. E. and Ruth, U.: A new Greenland ice core chronology for the last glacial termination, Journal of Geophysical Research: Atmospheres, 111(D6), 1–16, doi:10.1029/2005JD006079, 2006.

Reimer, P., Bard, E., Bayliss, A., Beck, J., Blackwell, P., Ramsey, C., Buck, C., Cheng, H., Edwards, R., Friedrich, M., Grootes, P., Guilderson, T., Haflidason, H., Hajdas, I., Hatté, C., Heaton, T., Hoffmann, D., Hogg, A., Hughen, K., Kaiser, K., Kromer, B., Manning, S., Niu, M., Reimer, R., Richards, D., Scott, E., Southon, J., Staff, R.,

Turney, C. and van der Plicht, J.: Intcal13 and Marine13 radiocarbon age calibration curves 0–50,000 years cal BP, Radiocarbon, 55(4), 1869–1887, doi:10.2458/rc.v51i4.3569, 2013.

Tarasov, P. E., Müller, S., Zech, M., Andreeva, D., Diekmann, B. and Leipe, C.: Last glacial vegetation reconstructions in the extreme-continental eastern Asia: Potentials of pollen and n-alkane biomarker analyses, Quaternary International, 290–291, 253–263, doi:10.1016/j.quaint.2012.04.007, 2013.

Tuthorn, M., Zech, R., Ruppenthal, M., Oelmann, Y., Kahmen, A., del Valle, H. F., Eglinton, T., Rozanski, K. and Zech, M.: Coupling $\delta^2$H and $\delta^{18}$O biomarker results yields information on relative humidity and isotopic composition of precipitation - a climate transect validation study, Biogeosciences, 12, 3913–3924, doi:10.5194/bg-12-3913-2015, 2015.

---

## Author Response (AR1)

**Reply to editorial decision letter and reviewer comments**

Dear Editor, dear Keely,

we are very grateful for your great editorial support and your positive editorial decision. Readily we follow your kind invitation to submit a revised version of our manuscript. As recommended by the reviewers, whom we once again cordially thank for their constructive feedback and suggestions, and you, we particularly tried to shorten the MS (reduction from almost 18,000 to less than 13,500 words). We furthermore included clarifications where necessary. Please find a point-by-point reply to your editorial decision letter below.

With best wishes,
Johannes Hepp & Michael Zech (on the behalf of all co-authors)

*Reviewer #1: please address all comments included here.*

→ We fully agree with Referee #1 that temperature is known to have a strong effect on $\delta^2H/\delta^{18}O_{precipitation}$ (and thus as well on $\delta^2H_{n\text{-}alkane}/\delta^{18}O_{sugar}$). Unlike originally planned (see our reply to Referee #1), we however prefer not to include the reconstructed $\delta^2H/\delta^{18}O_{source\text{-}water}$ record for the Gemündener Maar. Rather, in order not to extend our MS, we simply included the temperature effect now already in the abstract as well as in the revised Fig. 5. Please let us know in case you do not agree with this suggestion.

→ Following the recommendation of Referee #1 we explain 'LEL' earlier in the revised MS and shortened the MS (from almost 18,000 to less than 13,500 words).

*SC1: Clarification of the chronology should be included in the revision, however, if this will only add length to the manuscript, I would encourage you to submit a supplementary file that clearly articulates (and graphically represents?) how your chronology was derived. The point regarding coring location is easily clarified in your sites/methods. With regards to cut off for auto/alloch inputs for C/N values, I too would agree with the use of 10.*

→ We completely revised the age-depth model as suggested by B. Zolitschka in SC1 and written in our reply. However, we realized this without adding length to the manuscript. We divided the former chapter '2.2 Bulk analysis and age control' into '2.2 Bulk analysis and pollen analysis' and '2.3 Age control'. We also added one figure (new Fig. 2, in combination with new Fig. 3) to the manuscript in order to better illustrate how the age-depth model is derived. Therefore, we did not add a supplementary data file to explain the age-depth model. The coring location and some explanation about the location we added in chapter 2.1.

→ Readily we refer to the C/N threshold of 10 according to Meyers (2003) in our revised MS.

*Reviewer #2: Again, mostly relates to reducing bulk, being more succinct and clear throughout.*

→ We are very grateful to P.A. Meyers (Referee #2) for his constructive and encouraging comments. Following his recommendation, we shortened and partly rewrote our MS in order to make it more succinct and clear.

*SC2: Please include reference in your revision to d-excess, adding in some of the material cited in response (if not already in paper)*

→ Readily included in the first paragraph of chapter 3.3

*SC3: Again, adding in some of the material included here in a more succinct format would be useful. I realise this reviewer is the least supportive of the work, but some of the other comments you need to address may go some way to reconciling this review. Of course, this review will continue to be available for transparency.*

→ We are very grateful for your careful and balanced assessment of SC3. Indeed, Kahmen et al. (2013), where E. Schefuß is co-author, is already cited and discussed thoroughly in terms of dicotyledonous versus monocotyledonous in chapter 3.2.

*SC4: Regarding your statement of encouraging the modelling community to use this work in the absence of palaeoclimate modelling as part of this paper - please can you make your data open access and include a link to it (if you have not already done so). I am unsure why you would remove the statistics you have when asked for more? Are there additional analyses that could be undertaken to address the issues raised? If not, rather than remove, please clarify and comment in the manuscript. In the absence of additional analyses, please just ensure that your text does not suggest that correlation = causation, rather be explicit about the similarities and the fact this in itself lends to further investigation and (may) be helped by use of palaeoclimate modelling scenarios.*

→ Readily we follow your recommendation and include our data in a supplementary excel file in order to make it available for the modelling community.

→ We prefer to remove the Monte Carlo simulation. In our opinion, further statistical tests and analyses are not essential for the take home message of our MS. We readily follow your recommendation and carefully revised our MS in order to avoid a false correlation = causation interpretation.

References

[revised manuscript text omitted]
:  (black line) = no correction for grasses,   dₗ*  (light blue line) = δ²H corrected for grasses,   dₗ#  (light red line) = δ¹⁸O corrected for grasses,  dₗ#*  (light green line) = δ²H and δ¹⁸O corrected for grasses. The error bars of dₗ values are calculated according to Eq. (7). (B) Reconstructed RHₐᵥ records .

Modern RH variability during daytime and vegetation period (RH$_{dv}$) is displayed as boxplot derived from the adjacent meteorological station Nürburg-Barweiler, using monthly means from April to October between 6 a.m. and 7 p.m. (based on hourly data from 1995 to 2015; Deutscher Wetterdienst, 2016). The bold numbers within the boxplot represent the maximum, median, and minimum value, respectively. (C) Depth profile of *Poaceae* pollen . Additionally, the resampled data points (black points) are displayed. The GM1 core picture with the used age markers are displayed on the left. AL = Allerød, LST = Laacher See Tephra, YD = Younger Dryas, PB = Preboreal, and BO = Boreal.

[Figure]

**Fig. 67**: (A) Reconstructed  Gemündener Maar (GM) RH$_{dv}$ record. The bold line shows the three point moving average. Error bars and the blue shaded area indicate analytical uncertainties  calculated according to error propagation (Eq. 7).

~~lines represents the adjusted ages according to the Monte-Carlo-Simulation based correlation procedure to the IntCal13 $^{14}$C production rate. The GM RH$_{dv}$ values are additionally shown on an optimized age-depth scale (thin black line with squares). Please note that the $^{14}$C charcoal age is also covered by an error and was as well included in the adjustment procedure but not shown in this graph for clarity reasons.9,80100~~200 a BP). RAPiD-12-1K and RAPiD-15-4P *G. bulloides* and *G. inflat* data from Thornalley et al. (2009) and Thornalley et al. (2010), respectively. RAPiD-15-4P *N. pachyderma* data from Thornalley et al. (2011). Note that each record is plotted on its own timescale (planktonic Mg/Ca data see (Thornalley et al., 2009, 2010), $^{10}$Be data on GICC05 (Rasmussen et al., 2006), $^{14}$C data on IntCal13 calibration curve (Reimer et al., 2013) and RH$_{dv}$ data on Lake GM age-depth model, see Fig. 2C). AL = Allerød, LST = Laacher See Tephra, YD = Younger Dryas, PB = Preboreal and BO = Boreal.

**Table**

~~Tab. 1: Scenarios (1-4) comprising the different reconstructed leaf water parameters based on the *n*-alkane/sugar biomarker, the respective equations used for this reconstruction, the resulting deuterium-excess of leaf water based on the reconstructed leaf water as input for Eq. (5) and the finally derived relative air humidity values during daytime and vegetation period (RH$_{dv}$) with this equation.~~

**Tab. 1:** Scenarios 1-4 used for reconstructing deuterium-(d-)excess of leaf water and corresponding RH$_{dv}$ values in order to assess/estimate the effect of variable grass contributions on the reconstructed Gemündener Maar RH record (see also Fig. 6).

| scenario | leaf water reconstructed from *n*-alkane/sugar biomarkers | equations used for leaf water reconstruction | resulting -excess of leaf water as input for Eq. (5) | relative air humidity during daytime and vegetation period according Eq. (5) |
|---|---|---|---|---|
| 1 | $\delta^2H_l/\delta^{18}O_l$ | (8) and (9) | $d_l$ | RH$_{dv}$ |
| 2 | $\delta^2H_l*/\delta^{18}O_l$ | (8) and (9) + (11) | $d_l*$ | RH$_{dv}*$ |
| 3 | $\delta^2H_l/\delta^{18}O_l^{\#}$ | (8) + (10) and (9) | $d_l^{\#}$ | RH$_{dv}^{\#}$ |
| 4 | $\delta^2H_l*/\delta^{18}O_l^{\#}$ | (8) + (10) and (9) + (11) | $d_l*^{\#}$ | RH$_{dv}*^{\#}$ |

---

## Author Response (AR2)

**Reply to editorial decision letter and reviewer comments**

Dear Editor, dear Keely,

we are very grateful for your great editorial support and your positive editorial decision. Readily we follow your kind invitation to submit a revised version of our manuscript. Please find a point-by-point reply to your editorial decision letter below.

With best wishes,
Johannes & Michael & Co-authors

*Reviewer #2*

*Some minor rough areas remain in the manuscript that the authors need to address. For one, the accepted symbol for deuterium is "D", not "d" as the authors seem to use throughout out the text and the figure legends.*

→ In the current manuscript we use the common notion $\delta^{18}O$ for $^{18}O/^{16}O$, $\delta^2H$ for $^2H/^1H$ and d for the deuterium-excess, following the definition given by Dansgaard (1964).

*For another, a contradiction seems to exist between lines 17-19 on page 18 that state that atmospheric circulation over Europe has not changed much over the past 35 ky and lines 7-12 on page 19 that invoke changes in the Westerlies during the Younger Dryas. This seeming contradiction needs clarification.*

→ Thank you very much for pointing us to this discrepancy. We changed the sentence on page 18 to: "In addition, paleowater samples from Europe suggest that the d-excess of precipitation was rather constant throughout the past 35,000 years, which implies that the principle atmospheric circulation patterns over the European continent did not change substantially (Rozanski, 1985).". On page 18 we focus on long-term d-excess changes and argue that d-excess in precipitation was rather stable over the last 35,000 years (Rozanski, 1985), which implies that the principle water vapour transport patterns providing moisture for the European continent did not change much. However, shifts as well as strengthening or weakening of e.g. the Westerlies was still happening (see literature on page 19).

*A third issue is that the TOC/N ratios cited on lines 19-23 are atomic ratios, not weight ratios. Atomic ratios are 1.15 times larger than weight ratios, which is not much but is enough to influence their interpretations as being aquatic vs land plant indicators.*

→ Thank you very much for pointing us to this issue. Following your suggestion, we use TOC/N atomic ratios rather than TOC/N mass ratios throughout the revised manuscript. This is now also clearly stated in the text as well as in Fig. 3. Please note that our interpretation is not influenced by the change from mass to atomic TOC/N ratios.

*In addition, a number of minor syntactical corrections should be considered:*
*Page 2, line 1 – change to read "Causes of the Late Glacial to"*

→ Corrected.

*Page 2, line 4 – replace "crucial" by "keys"*

→ Corrected.

*Page 3, line 7 – change to read "Explanation for the Younger Dryas"*

→ Corrected.

*Page 3, line 12 – change to read "reconstruction by providing"*

→ Corrected.

*Page 3, line 17 – replace "eaf" by "leaf"*

→ Corrected.

*Page 3, line 28 – change to read "during recent years"*

→ Corrected.

*Page 7, line 6 – replace "grinded" by "ground"*

→ Corrected.

*Page 11, line 3 – replace "nominator" by "numerator"*

→ Corrected.

*Page 11, line 28 – specify whether the TOC/N ratios are atomic or weight here and in Figure 3 legend*

→ See reply above, we calculated and show now in the manuscript atomic ratios.

*Page 12, line 10 – replace "more positive" by "less negative"*

→ Corrected.

*Page 12, line 17 – replace "autochthon" by "authochthonous" and "allochthon" by "allochthonous"*

→ Corrected.

*Page 13, line 25 – change "significant" to "significantly"*

→ Corrected.

*Page 14, line 12 – change "composition" to "compositions"*

→ Corrected.

*Page 15, line 3 – replace "requested" by "required"*

→ Corrected.

*Page 15, line 30 – change "imply" to "implies"*

→ Corrected.

*Page 21, line23 – The phrase "which triggered solar insolation" is confusing. Insolation is a fancy word for solar heating. Solar activity certainly is responsible for solar heating, but what does Nort*h Atlantic temperature have to do with causing it? Please clarify.

→ Changed to: "We propose that both the North Atlantic Ocean temperature and solar activity (the latter triggering solar insolation) were the two main drivers for the RHdv variability in Central Europe.".